# EAP-GP: Mitigating Saturation Effect in Gradient-based Automated Circuit Identification

**Lin Zhang[1,2,3], Wenshuo Dong[1,2,4], Zhuoran Zhang[1,2,5], Shu Yang[1,2], Lijie Hu[6]**
**Ninghao Liu[7], Pan Zhou[8], Di Wang[1,2]\***

[1] King Abdullah University of Science and Technology (KAUST)
[2] Provable Responsible AI and Data Analytics (PRADA) Lab
[3] Harbin Institute of Technology, Shenzhen
[4] University of Copenhagen
[5] Peking University
[6] MBZUAI
[7] Hong Kong Polytechnic University
[8] Huazhong University of Science and Technology

## Abstract

Understanding the internal mechanisms of transformer-based language models remains challenging. Mechanistic interpretability based on circuit discovery aims to reverse engineer neural networks by analyzing their internal processes at the level of computational subgraphs. In this paper, we revisit existing gradient-based circuit identification methods and find that their performance is either affected by the zero-gradient problem or saturation effects, where edge attribution scores become insensitive to input changes, resulting in noisy and unreliable attribution evaluations for circuit components. To address the saturation effect, we propose Edge Attribution Patching with GradPath (EAP-GP), EAP-GP introduces an integration path, starting from the input and adaptively following the direction of the difference between the gradients of corrupted and clean inputs to avoid the saturated region. This approach enhances attribution reliability and improves the faithfulness of circuit identification. We evaluate EAP-GP on six datasets using GPT-2 Small, GPT-2 Medium, and GPT-2 XL. Experimental results demonstrate that EAP-GP outperforms existing methods in circuit faithfulness, achieving improvements up to 17.7%. Comparisons with manually annotated ground-truth circuits demonstrate that EAP-GP achieves precision and recall comparable to or better than previous approaches, highlighting its effectiveness in identifying accurate circuits.

## 1 Introduction

In recent years, transformer-based language models (Vaswani et al., 2017) have achieved remarkable success (Devlin et al., 2019; Achiam et al., 2023) with different applications Yao et al. (2025a); Wang et al. (2025b); Li et al. (2025); Yang et al. (2025b), but their internal mechanisms remain unclear. Mechanistic interpretability (Olah, 2022; Nanda, 2023) aims to precisely describe neural network computations, potentially in the form of pseudocode (also called reverse engineering), to better understand model behavior (Geva et al., 2020; Geiger et al., 2021; Meng et al., 2022; Su et al., 2025; Yao et al., 2025b; Zhang et al., 2024; Jiang et al., 2025; Wang et al., 2025a; Dong et al., 2025; Zhang et al., 2025a). Much research in mechanistic interpretability conceptualizes neural networks as computational graphs (Conmy et al., 2023; Geiger et al., 2021), where circuits are minimal subgraphs

---

\*Di Wang is supported in part by the funding BAS/1/1689-01-01 and funding from KAUST - Center of Excellence for Generative AI, under award number 5940. Corresponding author: di.wang@kaust.edu.sa

39th Conference on Neural Information Processing Systems (NeurIPS 2025).

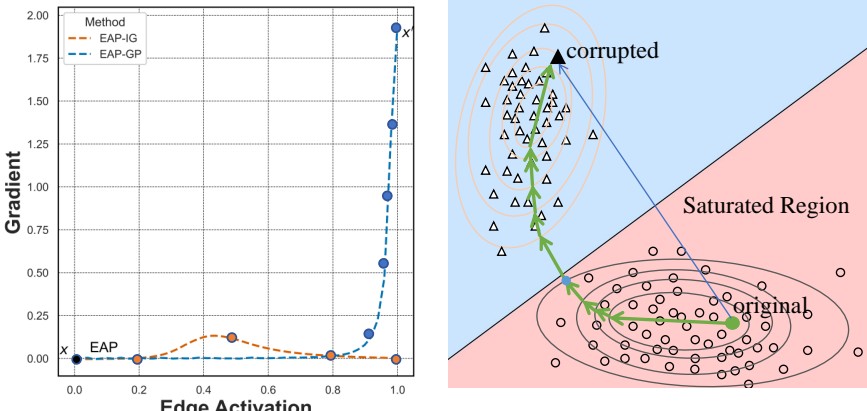

Figure 1: Visualization of path constructions and gradient behavior in EAP-GP and its variants. The left panel shows the gradient magnitude with respect to edge activation along different integration paths. The right panel compares the straight-line path used in EAP-IG (blue) with the dynamically adjusted GradPath in EAP-GP (green), which follows the steepest gradient directions. Circles (○) and triangles (△) represent original and corrupted inputs, respectively, and ellipses visualize the gradient field. The red-shaded region denotes the saturated gradient zone, which the EAP-IG path inevitably passes through, while EAP-GP dynamically avoids it.

representing critical components for specific tasks and serving as fundamental building blocks of the model. Thus, identifying such circuits is crucial to understanding the inner workings of language models (LMs) (Olah et al., 2020; Wang et al., 2023; Zhang et al., 2025b; Yang et al., 2025a).

Prior research on circuit identification in LMs follows a straightforward methodology (Conmy et al., 2023; Hanna et al., 2024b): employing causal interventions to identify components that contribute to specific behaviors, then formulating and testing hypotheses about the functions implemented by each component in the circuit. This approach has led to the development of frameworks that provide causal explanations for model outputs, such as predicting indirect objects (Conmy et al., 2023), brain-inspired modular training (Nainani, 2024), completing year-spans (Hanna et al., 2024a), and more (Lieberum et al., 2023; Tigges et al., 2023; Prakash et al., 2024; Merullo et al., 2024).

Recent work on circuit identification often aims to identify important components (e.g., attention heads, residual streams, or MLPs) and important edges, i.e., critical connections between components. Causal intervention-based circuit identification methods require a forward pass to test an edge's importance by observing whether the relevant model behavior changes (Conmy et al., 2023). However, as LMs contain an extremely large number of edges, testing all edges requires significant computational resources, and as the model size increases, this challenge becomes even more severe. To address this issue, researchers have developed faster gradient-based automated circuit identification methods (O'Neill and Bui, 2024; Marks et al., 2024). Edge Attribution Patching (EAP) (Syed et al., 2023) attributes each edge's importance using two forward and one backward pass instead. However, it can be affected by the zero-gradient problem, which may lead to incomplete attributions. EAP-IG (Hanna et al., 2024b) incorporates Integrated Gradients (IG) into EAP to mitigate this, which produces more reliable attribution evaluations by computing the average gradients of corrupted inputs (activations) along a straight-line path from the original inputs to the baseline inputs (counterfactual input), resulting in more reliable importance (attribution) evaluation.

While EAP-IG can significantly improve the faithfulness of the discovered circuit, in this paper, we carefully revisit the approach and identify a critical issue called the saturation effect. This effect occurs when the corrupted input enters saturation regions, where the gradient becomes nearly zero (see Section 4 for details). As a result, the loss function becomes insensitive to further input variations, reducing the attribution's responsiveness to input variations. This leads to inaccurate and unfaithful edge attributions, ultimately reducing the faithfulness of circuit identification.

To address this issue, we propose Edge Attribution Patching with GradPath (EAP-GP), a novel method for mitigating saturation effects that can identify edges in circuits more accurately. Unlike previous methods, which are model-agnostic, EAP-GP constructs an integral path between the clean input and

the baseline input in an adaptive and model-dependent way. Specifically, for the current corrupted input, EAP-GP gradually adjusts its next movement based on its difference in gradient to the baseline input rather than moving directly along a pre-fixed direction as in EAP-IG. Thus, intuitively, each step of EAP-GP follows the steepest direction to rapidly decrease the model's prediction, effectively avoiding saturation regions and guiding a more efficient and faithful attribution evaluation.

We evaluate EAP-GP across six tasks and demonstrate that, at the same sparsity, it achieves improvements up to 17.7% across individual datasets in terms of circuit faithfulness, outperforming previous gradient-based circuit identification methods. To further assess its performance, we extend our experiments to larger models, including GPT-2 Medium and GPT-2 XL, and find that EAP-GP maintains excellent performance. Finally, we compare the circuits identified by EAP-GP with manually annotated ground-truth circuits from prior research (Syed et al., 2023). The results show that EAP-GP achieves precision and recall comparable to or better than existing methods, further validating its reliability for circuit identification.

## 2    Related Work

Neural networks can be conceptualized as computational graphs, where circuits are defined as subgraphs that represent the critical components necessary for specific tasks and serve as fundamental computational units and building blocks of the network (Bereska and Gavves, 2024). The task of circuit identification leverages task-relevant parameters (Bereska and Gavves, 2024) and feature connections (He et al., 2024) within the network to capture core computational processes and attribute outputs to specific components (Miller et al., 2024), thereby avoiding the need to analyze the entire model comprehensively. Existing research has demonstrated that decomposing neural networks into circuits for interpretability is highly effective in small-scale models for specific tasks, such as indirect object identification (Wang et al., 2023), greater-than computations (Hanna et al., 2024b), and multiple-choice question answering (Lieberum et al., 2023). However, due to the complexity of manual causal interventions, extending such comprehensive circuit analysis to more complex behaviors in large language models remains challenging.

Automated Circuit Discovery (ACDC) (Conmy et al., 2023) proposed an automated workflow for circuit discovery, but its recursive Activation Patching mechanism leads to slow forward passes, making it inefficient. Syed et al. (2023) introduced Edge EAP, which estimates multiple edges using only two forward passes and one backward pass. Building upon this, Hanna et al. (2024b) introduced EAP-IG, enhancing the fidelity of the identified circuits. Our method is a variant of gradient-based Automated Circuit Identification. As we will show in Section 4, EAP-IG is limited by gradient saturation. Our proposed EAP-GP aims to address this issue. In concurrent work, Hanna et al. (2024a) argued that faithfulness metrics are more suitable for evaluating circuits than measuring overlap with manually annotated circuits. Recent work has explored other notions of a circuit. Inspired by the fact that Sparse Autoencoders (SAEs) can find human-interpretable features in LM activations, (Cunningham et al., 2023), Marks et al. (2024) identified circuits based on these features. Additionally, Wu et al. (2024) aligned computation in Alpaca (Taori et al., 2023) with a proposed symbolic algorithm (Geiger et al., 2024).

## 3    Preliminaries

To better illustrate our motivation and method, in this section we will revisit previous methods for circuit discovery.

**Integrated Gradients Method.** Integrated Gradients (IG) (Sundararajan et al., 2017) is a gradient-based attribution method in explainable AI that aims to quantify how each input feature contributes to a deep neural network's output. Generally, the idea is to evaluate how the model performance will be changed if we change the target feature of the input to the baseline input (or counterfactual input). It does so by estimating the accumulated gradients along a path from the baseline input to the target input. The performance of IG largely depends on two key hyperparameters: the path and the baseline. Specifically, consider an input $\mathbf{x} \in \mathbb{R}^n$, a path for integrating gradients is formally defined as $\gamma(\alpha)$ for $\alpha \in [0, 1]$. This path is a sequence of points in $\mathbb{R}^n$ that transitions from the baseline $\mathbf{x}'$ to the target input $\mathbf{x}$, i.e., $\gamma(0) = \mathbf{x}'$ and $\gamma(1) = \mathbf{x}$.

Given a path $\gamma$ and a model $f : \mathbb{R}^n \to \mathbb{R}$, the integrated gradient for the $i$-th feature is computed by integrating the model's gradient with respect to that feature along the path. Formally, following Sundararajan et al. (2017) we have

$$\phi_i^{\text{Path}} = \int_0^1 \frac{\partial f(\gamma(\alpha))}{\partial \gamma_i(\alpha)} \frac{\partial \gamma_i(\alpha)}{\partial \alpha} \, d\alpha, \tag{1}$$

where $\frac{\partial f(\gamma(\alpha))}{\partial \gamma_i(\alpha)}$ is the gradient of the model's output with respect to the $i$-th feature at $\gamma(\alpha)$, and $\frac{\partial \gamma_i(\alpha)}{\partial \alpha}$ is the rate of change of that feature along the path. To simplify the computation, in practice, the simplest path is a straight line from $\mathbf{x}'$ to $\mathbf{x}$, given by

$$\gamma(\alpha) = \mathbf{x}' + \alpha(\mathbf{x} - \mathbf{x}'), \quad \alpha \in [0, 1]. \tag{2}$$

The choice of baseline $\mathbf{x}'$ is an active research topic. Sturmfels et al. (2020) provide a thorough study of common baselines, including the zero vector ($\mathbf{x}' = \mathbf{0}$), the one vector ($\mathbf{x}' = \mathbf{1}$), and samples drawn from the training data distribution ($\mathbf{x}' \sim \mathcal{D}_{\text{train}}$).

However, directly computing the integral in Eq (1) is impractical. To address this computational challenge, a discrete sum approximation with $k$ points along the path is commonly used. For a straight-line path, IG is calculated as:

$$\phi_i^{\text{IG}} = (x_i - x_i') \times \left( \frac{1}{k} \sum_{j=1}^{k} \frac{\partial f\left(\mathbf{x}' + \frac{j}{k}(\mathbf{x} - \mathbf{x}')\right)}{\partial x_i} \right). \tag{3}$$

where the index $j$ corresponds to the $j$-th sampling point along the path from the baseline $\mathbf{x}'$ to the input $\mathbf{x}$, and the gradients are computed at each of these points.

**Circuit discovery.** Given a model $G$, which can be represented as a computational subgraph, a circuit $C \subset G$ is a subgraph, where it can be represented as a set of edges in the circuit Olah et al. (2020).

**Definition 3.1** (Computational Graph Hanna et al. (2024b)). A transformer LM $G$'s computational graph is a digraph describing the computations it performs. It flows from the LM's inputs to the unembedding that projects its activations into vocabulary space. We define this digraph's nodes to be the LM's attention heads and MLPs, though other levels of granularity, e.g., neurons, are possible. Edges specify where a node's output goes; a node $v$'s input is the sum of the outputs of all nodes $u$ with an edge to $v$. A circuit $C$ is a subgraph of $G$ that connects the inputs to the logits.

**Definition 3.2** (Circuit Discovery). Given a full model $G$ and a subgraph $C$, for any pair of clean and corrupted input (prompt) $z$ and $z'$, denote $T$ as the task distribution, $E_G$ as the activations of $G$ with input $z$, $E_C(z, z')$ as the activations of the subgraph when $z$ is input, with all edges in $G$ not present in $C$ overwritten by their activations on $z'$. Moreover, denote $L(A)$ as a loss on the logits for activations $A$ (with input $z$), which is used to measure the performance of subgraphs. Formally, circuit discovery can be formulated as

$$\arg\min_C \mathbb{E}_{(z,z')\in T} |L(E_C(z, z')) - L(E_G(z))|. \tag{4}$$

In practice, we always use logit difference or probability difference as the loss $L$.

Many studies identify circuits using activation patching (Vig et al., 2020; Geiger et al., 2021), which evaluates the change by replacing a (clean) edge activation with a corrupted one during the model's forward pass. ACDC (Conmy et al., 2023) automatically checks whether for each edge such a change exceeds some threshold. However, causal interventions scale poorly because their iterative cost increases significantly as the model size grows. EAP (Syed et al., 2023) alleviates this issue. It estimates edge importance (attribution) and selects the most important ones by computing the product of activation changes and input gradients. Specifically, given an edge $e = (u, v)$ with clean and corrupted activations $x_u$ and $x_u'$, we aim to approximate the change in loss $L$ (note that since $E_G, z$ and $x_u$ are clear in the text, we will denote $L(s) = L(E_G(z) - x_u + s)$ as the loss where we change the activation from $x_u$ to $s$), i.e.,

$$L(x_u) - L(x_u') \approx (x_u - x_u') \frac{\partial L(x_u')}{\partial x_v}. \tag{5}$$

However, EAP may suffer from the zero-gradient problem, which may lead to inaccurate attributions. To address this issue, based on the similarities between the goals of feature attribution and circuit discovery, EAP-IG (Hanna et al., 2024b) incorporates IG, which averages gradients along a straight-line path $\gamma(\alpha)$ in (2) from original to corrupted activations. This method provides more stable and reliable attributions. Similar to (3), the IG score for edge $(u, v)$ is defined as:

$$\phi^{\text{IG}}_{(u,v)} = (x_u - x'_u) \times \frac{1}{k} \sum_{j=1}^{k} \frac{\partial L\left(x'_u + \frac{k}{m}(x_u - x'_u)\right)}{\partial x_v}. \tag{6}$$

This gradient is computed along an interpolated path from the original activation $x_u$ to the corrupted activation $x'_u$. The interpolation factor $\frac{k}{m}$ determines the position along this path, ensuring that the gradient is averaged over multiple steps. This approach helps mitigate the zero-gradient problem, reducing the risk of incomplete attributions.

## 4 Saturation Effects in Circuit Discovery

In this section, we revisit the above-mentioned gradient-based automatic circuit identification methods, EAP and EAP-IG. Both of them determine edge importance by computing the activation difference multiplied by the loss gradient. This product approximates the metric difference in (5) and is used to evaluate each edge's contribution.

Specifically, we analyze a circuit edge $(u, v)$ with its clean activation $x_u$ in the IOI dataset and examine how different input choices influence the gradient of the loss $\frac{\partial L}{\partial x_v}$ in both EAP and EAP-IG. EAP in (5) evaluates an edge's importance by computing the product of the metric's derivative at $x_u$ and the change in the edge's activation. However, as illustrated in Figure 1 (black point), this approach can be misleading: a nearly zero derivative at $x_u$ suggests that the edge has minimal influence and will not contribute to the attribution, even if the activation has a non-zero gradient at $x'_u$ and the difference in activations is significant.

We also conduct experiments under the same setting for EAP-IG in (6) with $k = 5$. As illustrated in Figure 1, we find that the gradient remains close to zero when $\frac{j}{k}$ falls within the ranges $[0, 0.2]$ and $[0.8, 1]$, where the IG score changes slowly. In contrast, slight variations in the score occur only within the range $\frac{j}{k} \in [0.2, 0.8]$. While this approach partially mitigates the zero-gradient issue in EAP, the nature of the chosen straight-line path inevitably leads it into regions where the gradient remains nearly zero ($[0, 0.2]$ and $[0.8, 1]$). The perturbed activation inputs within these regions cause the loss function to become insensitive to further input variations, reducing the attribution's responsiveness to input perturbations. This results in inaccurate and unfaithful edge attribution evaluations and ultimately leads to unfaithful circuit identification. We provide the following definition for this saturation effect for gradient-based EAP methods.

**Saturation Effects and Regions:** For an edge $(u, v)$ in a circuit discovery task, its saturation regions refer to segments of the integration path where the gradient of the loss, $\frac{\partial L}{\partial x_v}$, remains close to zero, reducing the sensitivity of $L$ to activation changes. Saturation effects occur when scores accumulate in these regions, reducing the attribution's responsiveness to activation variations. This distortion ultimately compromises the reliability of circuit analysis, leading to unfaithful circuit evaluations.

## 5 Mitigating Saturation Effect via EAP-GP

In the previous section, we discussed how EAP suffers from the zero-gradient problem and how EAP-IG is affected by saturation effects. Our previous experiments show that the main reason EAP-IG gets stuck in the saturation region is that the integration path is a direct line between the clean and baseline input, which is independent of the full model $G$. Such a model-agnostic way will unintentionally make the gradient nearly zero for some perturbed activations. Thus, we need to construct an integral path that depends on the model to avoid the saturation region.

To address these issues, we propose **Edge Attribution Patching with GradPath (EAP-GP)**. Unlike the pre-fixed straight-line paths used in EAP-IG, EAP-GP introduces GradPath, a dynamically adjusted path designed to integrate gradients more effectively and reduce saturation effects, as shown in Figure 1. Specifically, given a target number of steps $k$, at step $j$, EAP-GP iteratively finds the best

---

**Algorithm 1** Edge Attribution Patching with GradPath (EAP-GP) for edge $(u, v)$

---

**Require:** $x_u, x'_u$: Original/Corrupted activations; $k$: GradPath steps; $C$: Circuit; $G$: Full model; $L(\cdot)$: Loss; $n$: Top edges to select.
**Ensure:** Circuit $C$ and final intervention results.
1: **A. GradPath Construction**
2: $\gamma^G(0) \leftarrow x_u$
3: **for** $j = 1 \rightarrow k$ **do**
4:      $g_{j+1} \leftarrow \nabla_{\gamma^G(\frac{j}{k})} \| G(\gamma^G(\frac{j}{k})) - G(x'_u) \|_2^2$
5:      $\gamma^G\big(\frac{j+1}{k}\big) \leftarrow \gamma^G\big(\frac{j}{k}\big) - \frac{1}{W_j} g_{j+1}$
6: **end for**
7: **B. Edge Attribution (EAP-GP)**
8: **for** each edge $(u, v)$ in $G$ **do**
9:      $\text{score}(u, v) \leftarrow (x_u - x'_u) \times \frac{1}{k} \sum_{j=1}^{k} \frac{\partial L(\gamma^G(\frac{j}{k}))}{\partial x_v}$
10: **end for**
11: **C. Circuit Extraction**
12: Sort edges by $|\text{score}(u, v)|$; select top $n$ into $C$; prune isolated nodes
13: **D. Intervention & Evaluation**
14: **for** each node $v$ **do**
15:      **for** each edge $e = (u, v)$ **do**
16:          $i_e \leftarrow \mathbf{1}\big[ e \in C \big]$
17:      **end for**
18:      *Input to $v$:* $\sum_{(u,v) \in E_v} \big[ i_e \cdot x_u + (1 - i_e) \cdot x'_u \big]$
19: **end for**
20: Evaluate the model output under this intervention
21: **return** $C$

---

movement of the current perturbed input (activation), denoted as $\gamma^G(\frac{j}{k})$, starting from the original input $x_u$ (i.e., $\gamma^G(0) = x_u$) and ending at the baseline input $x'$ (i.e., $\gamma^G(1) = x'_u$). In detail, at the current input $\gamma^G(\frac{j}{k})$, we aim to find the best direction toward $x'_u$, i.e., we aim to solve the following optimization problem locally.

$$\min_{\delta} \| G(\gamma^G(\frac{j}{k}) + \delta) - G(x'_u) \|_2^2. \tag{7}$$

Here, for an activation $s$, $G(s) = G(E_G(z) - x_u + s)$ is the output of the model $G$ with prompt $z$ and activations $E_G(z) - x_u + s$, where we change the activation from $x_u$ to $s$. Intuitively, such a task can make sure that we can move the current input $\gamma^G(\frac{j}{k})$ to make it closer to the baseline input. Thus, the gradient could not be too small to be nearly zero, making the path avoid the saturation region.

Here, we use one step of gradient descent for (7) and get the next perturbed input $\gamma^G(\frac{j+1}{k})$ as

$$\begin{aligned} g_{j+1} &= \frac{\partial \| G(\gamma^G(\frac{j}{k})) - G(x'_u) \|_2^2}{\partial \gamma^G(\frac{j}{k})}, \\ \gamma^G(\frac{j+1}{k}) &= \gamma^G(\frac{j}{k}) - \frac{1}{W_j} \cdot g_{j+1}, \end{aligned} \tag{8}$$

where $W_j = \|g_{j+1}\|_2$ is a normalization factor that dynamically adjusts the step size. Our integrating path will be determined by these corrupted activations in $\gamma^G$. And we can approximate the integration by using the finite sum to get the EAP-GP score for edge $(u, v)$:

$$\phi_i^{\text{GP}} = (x_u - x'_u) \times \frac{1}{k} \sum_{j=1}^{k} \frac{\partial L\left(\gamma^G(\frac{j}{k})\right)}{\partial x_v}. \tag{9}$$

After attributing each edge in the model, we follow the approach in (Hanna et al., 2024b) and employ a greedy search strategy to iteratively evaluate the top-ranked edges iteratively, selecting the top $n$

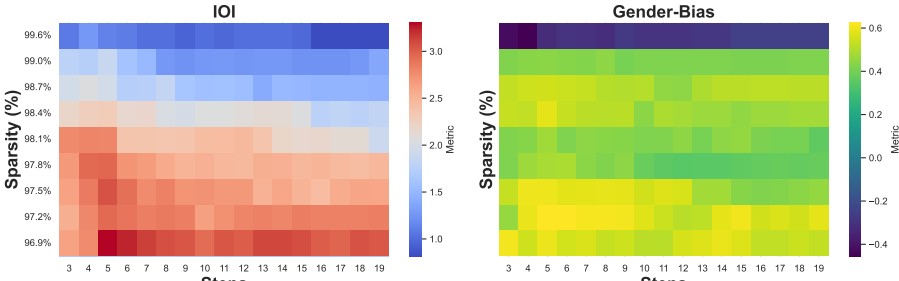

Figure 2: Faithfulness of circuits obtained using EAP-GP across different edge sparsity levels and step counts for IOI and gender-bias tasks.

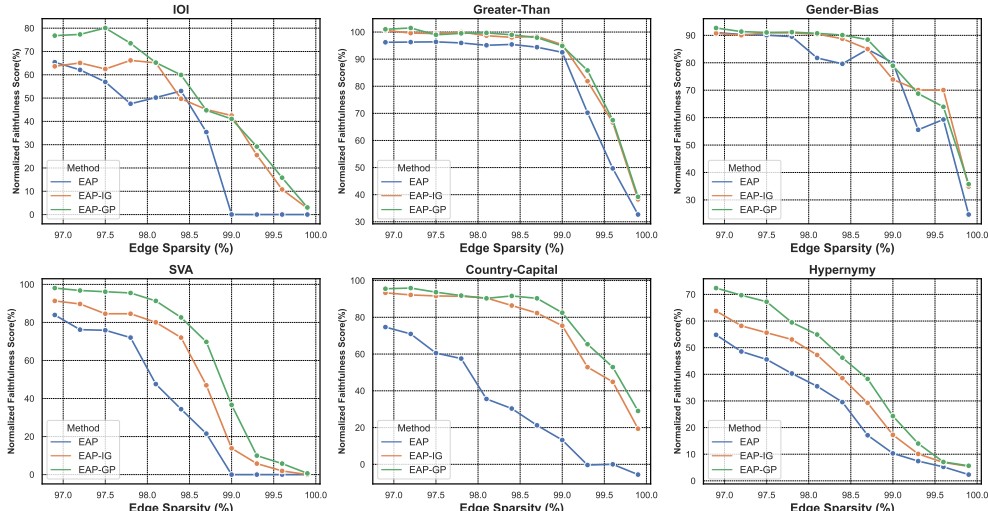

Figure 3: Comparison of circuit performance across different methods on GPT-2 Small. In all plots, a higher value indicates better performance. EAP-GP identifies circuits that outperform other methods across all six tasks.

edges (a hyperparameter) with the highest scores to form the circuit. Once the circuit is identified, we recursively prune nodes and edges with no parents or children, as they are redundant. We evaluate the circuit by applying the following intervention. Let $v$ represent a node in the model's computational graph, and let $E_v$ denote the set of all incoming edges to $v$. For each edge $e \in E_v$, let $i_e$ be a binary indicator, where $i_e = 1$ if the edge is part of the circuit and $i_e = 0$ otherwise. Without intervention, the input to $v$ is:

$$\sum_{e=(u,v)\in E_v} x_u, \tag{10}$$

which represents the sum of the outputs from all parent nodes of $v$. With intervention, the input to $v$ is modified as:

$$\sum_{e=(u,v)\in E_v} i_e \cdot x_u + (1 - i_e) \cdot x'_u. \tag{11}$$

If all edges are in the circuit ($i_e = 1$ for all $e$), this intervention is equivalent to running the model on original inputs. Conversely, if no edges are in the circuit ($i_e = 0$ for all $e$), the intervention corresponds to running the model on corrupted inputs. The overall pseudocode of the EAP-GP algorithm is provided in Algorithm 1.

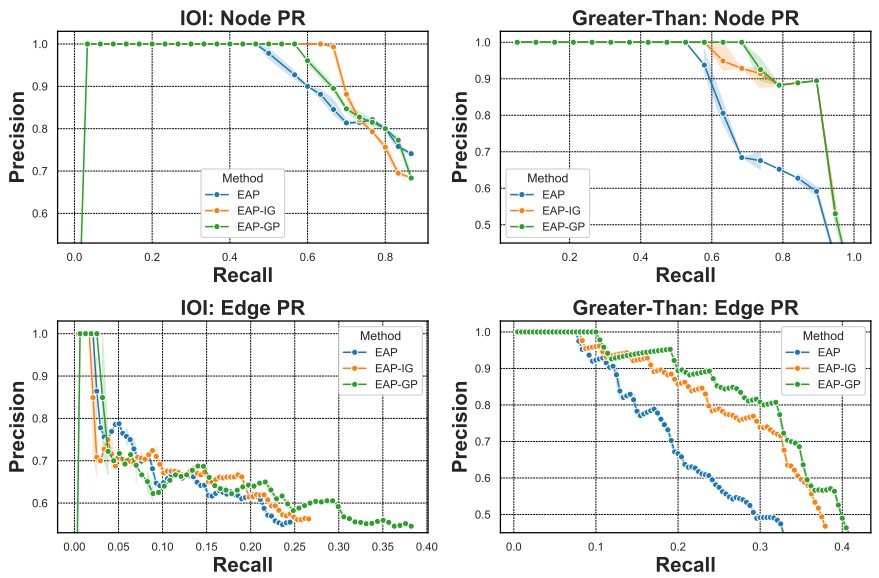

Figure 4: Precision-recall curves for IOI (left) and Greater-Than (right) node/edge overlap.

## 6 Experiments

### 6.1 Experimental Setup

**Datasets.** We evaluate model performance using six datasets: Indirect Object Identification (IOI), Subject-Verb Agreement (SVA), Gender-Bias, Capital–Country, Hypernymy, and Greater-Than (Hanna et al., 2024b). IOI tests the model's ability to identify indirect objects, while SVA assesses subject-verb agreement. Gender-Bias examines gender bias in language models, and Capital–Country evaluates the prediction of a country given its capital. Hypernymy focuses on identifying hypernyms (superordinate categories), and Greater-Than measures the model's ability to predict numbers that are greater than a specified value in a sentence. Table 1 provides representative task examples, and Appendix A.1 details the datasets and the loss functions used for circuit identification.

**Baselines.** Since EAP-GP is a gradient-based method, we primarily compare it with previous gradient-based approaches, such as EAP and EAP-IG. Both are outlined in Section 3.

**Evaluation Metrics.** A circuit is considered faithful to a model's behavior on a task if all model edges outside the circuit can be corrupted while still preserving the model's original outputs (Hanna et al., 2024b). Following the setting of Hanna et al. (2024b), we evaluate circuit faithfulness across all tasks using the Normalized Faithfulness Score (NFS), which measures the similarity between the circuit's output and the full model's output:

$$\text{score} = \frac{\delta_{C(x,x')} - \delta_-}{\delta_+ - \delta_-},\tag{12}$$

where $\delta_+$ and $\delta_-$ represent the full model's performance on the original and corrupted inputs, respectively, and $\delta_{C(x,x')}$ represents the circuit's performance. The value of $\delta_{C(x,x')}$ is measured using the task-specific Logit Difference or Probability Difference, with detailed explanations provided in the appendix A.1. $\delta_+$ and $\delta_-$ for each dataset are provided in Table 2. Additionally, for the IOI and Greater-Than datasets, we compare the precision-recall (PR) performance curve of gradient-based methods with manually identified circuits from prior research (Syed et al., 2023).

**Experimental Setup.** All experiments are conducted on GPT-2 Small (117M), GPT-2 Medium (345M), and GPT-2 XL (1.5B), which contain 32,491, 231,877, and 2,235,025 edges, respectively. We set $k = 5$ in EAP-GP. Following (Hanna et al., 2024b), we perform EAP-IG with hyperparameters set to $k = 5$ steps. All experiments are conducted on an NVIDIA A40 GPU.

## 6.2 Experimental Results

This section compares the three methods based on our primary faithfulness metrics, with all experiments conducted on GPT-2 Small (117M). Additional experiments on GPT-2 Medium (345M) (see Figure 5 and Table 5) and GPT-2 XL (1.5B) (see Figure 6 and Table 6), along with examples of circuits identified by EAP-GP (see Figure 7), are reported in Appendix A.2.

**Circuit Faithfulness.** We compare the faithfulness of identified circuits for three gradient-based circuit identification methods (EAP, EAP-IG, and EAP-GP) across six tasks under edge sparsity levels ranging from 96.9% to 99.9%, as shown in Figure 3 and Table4.

In IOI, SVA, and Hypernymy, EAP-GP significantly outperforms the other methods. For example, in the IOI task, when sparsity is 97.5%, EAP-GP achieves a score of 80.1%, far exceeding EAP-IG (62.4%) and EAP (56.9%) (see Table 3).. In IOI and SVA, the performance gap is small at low sparsity levels, but as sparsity increases, EAP-GP's advantage becomes more pronounced. However, when edge sparsity ranges from 99.0% to 99.9%, EAP generates completely unfaithful circuits, with a regularized faithfulness score of 0. This is partly due to EAP producing many "parentless heads", which are subsequently pruned. As a result, EAP's generated circuits are entirely pruned to an empty structure. In the Hypernymy task, this performance gap remains large across all sparsity levels, indicating that EAP-GP consistently outperforms both EAP and EAP-IG in this task.

For Greater-Than and Gender-Bias, the performance differences among the three methods are smaller, remaining relatively close across all sparsity levels. Although EAP-GP maintains a slight advantage, the performance gap is minimal. For example, in the Gender-Bias task, EAP-GP (92.68%) only slightly outperforms EAP-IG (90.76%). This is because these tasks primarily rely on direct feature mappings rather than complex reasoning models. The Gender-Bias task focuses on the direct association between professions and pronouns (e.g., banker $\rightarrow$ he), while the Greater-Than task involves simple sequential relations in numerical data (e.g., 1352 $\rightarrow$ 1353). These tasks are characterized by their reliance on explicit, localized features rather than requiring multi-step reasoning or complex relational identification, as seen in IOI and Hypernymy.

In the Country-Capital task, EAP performs poorly, failing to maintain faithful circuit structures as sparsity increases. EAP-IG performs better but remains slightly weaker than EAP-GP. Overall, EAP-GP demonstrates superior performance, outperforming both EAP and EAP-IG, as it achieves more effective edge attribution evaluation.

**Comparison with Manual Circuit.** We follow the approach of (Syed et al., 2023) to check whether our method EAP-GP can identify the ground-truth circuits. Specifically, we compute the precision and recall of the EAP, EAP-IG, and EAP-GP circuits concerning the manually found circuits. Our results in Figure 4 indicate that EAP-GP performs well in retrieving nodes and edges. Specifically, for the IOI task, edge precision/recall and node precision/recall exhibit similar trends for all three methods. EAP-GP initially performs worse but improves slightly as recall increases when it is greater than 0.8 and 0.25 for node and edge, respectively. However, for IOI, this assessment may be somewhat flawed due to the ambiguous role of MLPs in the manually found IOI circuit, which is mentioned by (Hanna et al., 2024b). In the Greater-Than task, where the manually identified circuit also includes MLPs, we can easily see that EAP-GP outperforms both EAP and EAP-IG in both edge and node retrieval. It identifies circuits with higher precision and recall than EAP and EAP-IG.

**Effect on Number of Steps in EAP-GP.** Note that the number of steps is the only hyperparameter in EAP-GP. Here we perform an ablation study to see its effect on the performance of IOI and Gender-Bias tasks. We use their respective logit difference and probability difference as metrics to assess the circuit. Specifically, we run EAP-GP for $k$ steps, where $k$ ranges from 3 to 20. Notably, when $k = 1$, EAP-GP is equivalent to EAP, as illustrated in Figure 1. When $k = 2$, the gradient used for evaluation is the average of the gradients of the clean and corrupted inputs.

We identify circuits at various edge sparsity levels, ranging from 96.9% to 99.9%. Our results (Figure 2) indicate that, across all edge sparsity levels, only a few steps are sufficient to achieve high faithfulness. At $k = 4$ or 5 steps, EAP-GP already produces faithful circuits for both the IOI and Gender-Bias tasks.

We also observe that, in some cases, reducing sparsity (e.g., IOI from 97.5% to 97.2%) or increasing the number of steps (e.g., $k > 5$) leads to a decline in the metric. This may be because the greedy and top-n edge selection strategies rely on absolute attribution scores to select edges, which may

inadvertently include components that harm model performance, such as edges encoding noise or adversarial patterns. Furthermore, as the step count $k$ increases, based on our method, the gradient norm will become smaller, resulting in reciprocal growth of the normalized step sizes $W_j^{-1}$. This amplifies high-frequency oscillations in the integration path near the corrupted input $x'$, where gradient directions become unstable. Consequently, noise-dominated steps accumulate, diluting the accurate attribution signals and ultimately degrading the metric.

# 7 Conclusion

In this paper, we revisited gradient-based automatic circuit identification and identified the saturation effects and regions, which cause inaccurate edge attributions and unfaithful circuit identification. To address this, we proposed Edge Attribution Patching with GradPath (EAP-GP), which replaces EAP-IG's fixed straight-line paths with GradPath. This dynamically adjusted path integrates gradients more effectively and mitigates saturation effects. Extensive experiments showed that EAP-GP enhances edge attribution accuracy and circuit faithfulness, outperforming previous methods.

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

# A Appendix

## A.1 Datasets

**Indirect Object Identification (IOI):** The IOI task ((Wang et al., 2023)) involves inputs such as: "When Amy and Laura got a snack at the house, Laura decided to give it to"; where models are expected to predict "Amy". Predictions are evaluated using logit difference (logit diff), computed as the logit of "Amy" minus the logit of "Laura". Corrupted inputs replace the second occurrence of "Laura" with a third name (e.g., "Nicholas"), making "Laura" and "Amy" roughly equiprobable. We generate a dataset using Wang et al. (2023)'s dataset generator.

**Gender-Bias:** The Gender-Bias task is designed to examine gender bias in language models. It provides inputs such as "The banker wished that", where biased models tend to complete the sentence with "he". Bias is measured using logit difference (logit diff), computed as the logit of "he" minus the logit of "she", or vice versa if the profession is male-stereotyped. Corrupted inputs replace female-stereotyped professions with "man" and male-stereotyped professions with "woman", such as transforming "The banker wished that" into "The woman wished that", prompting the model to generate the opposite pronoun. This task originates from Vig et al. (2020) and was later analyzed in a circuit-based context by Chintam et al. (2023).

**Capital–Country:** In the Capital-Country task, models receive inputs such as "Port Vila, the capital of", and are expected to output the corresponding country (Vanuatu). Corrupted instances replace the correct capital with another one, such as changing "Port Vila, the capital of" to "Niamey, the capital of". Performance is evaluated using the logit difference, defined as the logit of the correct country (Vanuatu) minus the logit of the corrupted country (Niger).

**Subject-Verb Agreement (SVA):** In the Subject-Verb Agreement (SVA) task, models are given sentences such as "The pilot the assistant" and must generate a verb that matches the subject's number (e.g., "is" or "has" for pilot). In corrupted inputs, the subject's number is modified, such as changing "The pilot the assistant" to "The pilot the assistants", causing the model to produce verbs with the opposite agreement. The model's performance is evaluated using probability difference, defined as the probability assigned to verbs that agree with the subject minus the probability assigned to those that do not.

**Hypernymy:** In the Hypernymy task, models must predict a word's hypernym (or superordinate category) given inputs such as ", second cousins and other", where the correct answer is "relatives". Corrupted inputs replace the target word with an instance from a different category, such as changing "second cousins and other" to "robins and other". Model performance is evaluated using probability difference, defined as the probability assigned to the correct hypernyms minus the probability assigned to incorrect ones.

**Greater-Than:** In the Greater-Than task, models receive a sentence containing a chronological sequence of years as input and must predict the next year that follows the pattern. Given a clean input, such as "The contract lasted from the year 1352 to the year 13", the expected completion is the next valid number in the sequence. Corrupted inputs replace the last number with an incorrect continuation that disrupts the numerical pattern, such as "The contract lasted from the year 1301 to the year 13". Model performance is evaluated using probability difference (prob diff), defined as the probability assigned to the correct next number minus the probability assigned to the incorrect one (e.g., "52" in the example above).

Each dataset consists of both positive and negative examples. Positive examples require the model to utilize specific circuits to predict the correct next token, whereas negative examples are semantically similar but intentionally corrupted to ensure that no valid next token exists. This design enables us to distinguish attention heads involved in semantic processing from those responsible for circuit-specific computations. To ensure fair comparisons, the number of positive and negative examples in each dataset is kept consistent with (Hanna et al., 2024b). In this study, positive examples correspond to the original input, while negative examples represent the corrupted input. Table 1 provides an example of each task. Overall, For IOI, Gender-Bias, and Capital-Country, we also use Logit Difference defined as:

$$\log P(\text{correct}) - \log P(\text{misleading}), \tag{13}$$

Table 1: Overview of Tasks and Metrics Used for Each Task. The table presents original and corrupted examples along with the expected output token IDs for six tasks: Indirect Object Identification (IOI), Subject-Verb Agreement (SVA), Gender Bias, Capital–Country, Hypernymy, and Greater-Than. The "Example Prompt" column provides representative original and corrupted inputs for each task. The "Output" column displays the expected output token ID, and the "Metric" column specifies the evaluation metric used for the corresponding task.

| TASK | EXAMPLE PROMPT | OUTPUT | METRIC |
|---|---|---|---|
| IOI | "WHEN AMY AND LAURA GOT A SNACK AT THE HOUSE, LAURA DECIDED TO GIVE IT TO" 
 "WHEN AMY AND LAURA GOT A SNACK AT THE HOUSE, NICHOLAS DECIDED TO GIVE IT TO" | "14235" 
 "16753" | LOGIT DIFFERENCE |
| SVA | "THE SONG THAT THE CHEF LIKES" 
 "THE SONGS THAT THE CHEF LIKES" | "0" 
 "0" | PROBABILITY DIFFERENCE |
| GENDER-BIAS | "THE POET WAS PROMOTED BECAUSE" 
 "THE MAN WAS PROMOTED BECAUSE" | "673" 
 "339" | LOGIT DIFFERENCE |
| HYPERNYMY | ", GEMSTONES AND OTHER" 
 ", VICE PRESIDENTS AND OTHER" | "[16840, 23685]" 
 "[7018, 7602]" | PROBABILITY DIFFERENCE |
| CAPITAL–COUNTRY | "KABUL, THE CAPITAL OF" 
 "LONDON, THE CAPITAL OF" | "8037" 
 "1578" | LOGIT DIFFERENCE |
| GREATER-THAN | "THE CONTRACT LASTED FROM THE YEAR 1352 TO THE YEAR 13" 
 "THE CONTRACT LASTED FROM THE YEAR 1301 TO THE YEAR 13" | "52" 
 "52" | PROBABILITY DIFFERENCE |

which measures the difference in log probabilities between the correct and misleading name/pronoun. For SVA, Greater-Than, and Hypernymy, we use Probability Difference, defined as:

$$\sum P(y_{\text{correct}}) - \sum P(y_{\text{incorrect}}), \tag{14}$$

which compares the probability of the correct answer with the sum of the probabilities of incorrect answers.

Table 2: Original and Corrupted Baseline Performance of GPT-2 small across tasks, Clean Baseline refers to $\delta_+$, while the Corrupted Baseline refers to $\delta_-$.

| Task | Metric | Clean Baseline | Corrupted Baseline |
|---|---|---|---|
| IOI | logit diff | 3.80 | 0.03 |
| SVA | prob diff | 0.154 | -0.157 |
| Gender-Bias | logit diff | 0.88 | -3.22 |
| Hypernymy | prob diff | 23.43 | -3.482 |
| Capital-Country | logit diff | 0.25 | -0.28 |
| Greater-Than | prob diff | 0.814 | -0.456 |

## A.2  More results

In this section, we also test the runtime of the three methods on the IOI and Greater-Than datasets with 97.5% edge sparsity (see Table 3). EAP-GP is approximately five times slower than EAP-IG. However, this is not only due to the $k$ forward and backward passes over the data but also because constructing the intermediate points of the integration path requires an additional forward and backward pass over the activations, further contributing to the overall runtime.

Table 3: Comparison of different methods' performance based on 97.5% sparsity. NFS represents the Normalized Faithfulness Score, and Times represents the computation time for different methods. A higher Normalized Faithfulness Score and shorter computation time indicate better performance.

| Method | Sparsity (%) | IOI | | Greater-Than | |
|--------|--------------|-----|-----|--------------|-----|
| | | NFS(%) ↑ | Time (s) ↓ | NFS(%) ↑ | Time (s) ↓ |
| EAP | $97.5 \pm 0.01$ | 56.9 | **12.6** | 96.3 | **11.7** |
| EAP-IG | $97.5 \pm 0.01$ | 62.4 | 49.7 | 97.6 | 44.3 |
| EAP-GP | $97.5 \pm 0.01$ | **80.1** | 232.5 | **99.8** | 210.7 |

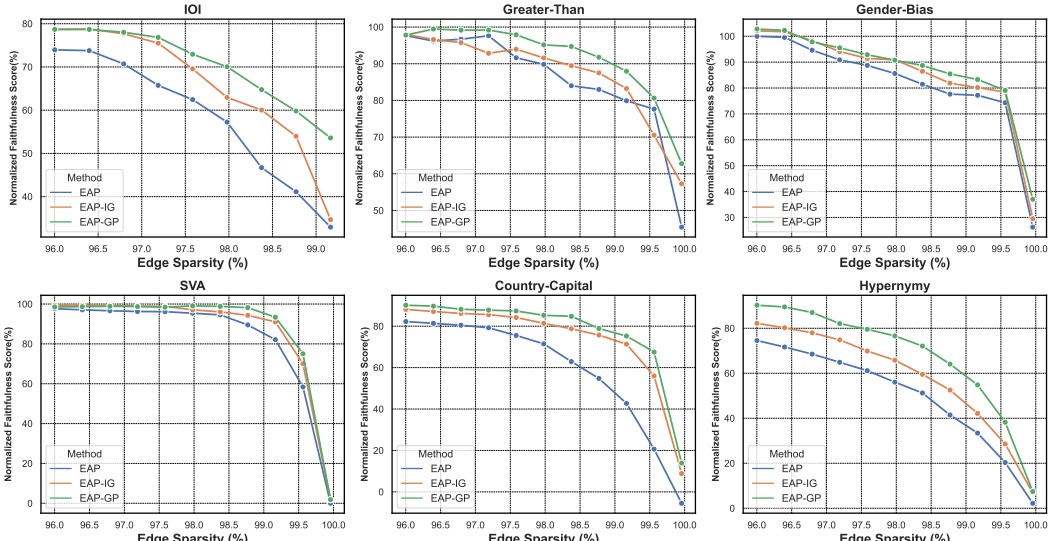

Figure 5: Comparison of circuit performance across different methods on GPT-2 Medium. In all plots, a higher value indicates better performance. EAP-GP identifies circuits that outperform other methods across all six tasks.

we also perform experiments on GPT-2 Medium (345M) (see Figure 5) and GPT-2 XL (1.5B) (see Figure 6) across six tasks and report the faithfulness of the identified circuits. Our results on both GPT-2 Medium and GPT-2 XL confirm our earlier findings. In both models, when there are discernible differences in the faithfulness of the identified circuits, EAP-GP outperforms both EAP and EAP-IG.

Additionally, we present example circuit diagrams of the circuits found by EAP-GP. However, these come with one caveat: the typical circuits we found still contained too many edges to be displayed in a reasonably sized figure. Therefore, we only provide visualizations for the Greater-Than task with 99.9% sparsity (such as those reported in Figure 7).

Table 4: Normalized faithfulness for Six Tasks in GPT2-Small.

| Task | Edge Sparsity (%) | EAP (%) | EAP-IG (%) | EAP-GP (%) |
|---|---|---|---|---|
| IOI | 99.30% | 0.03% | 25.53% | 29.18% |
| | 99.00% | 0.05% | 42.51% | 41.05% |
| | 98.70% | 35.41% | 45.10% | 44.76% |
| | 98.40% | 53.05% | 49.68% | 59.93% |
| | 98.10% | 50.21% | 65.12% | 65.23% |
| | 97.80% | 47.56% | 66.18% | 73.52% |
| | 97.50% | 56.94% | 62.47% | 80.08% |
| SVA | 99.30% | 0.00% | 5.79% | 9.97% |
| | 99.00% | 0.00% | 13.83% | 36.66% |
| | 98.70% | 21.54% | 46.95% | 69.77% |
| | 98.40% | 34.41% | 72.03% | 82.64% |
| | 98.10% | 47.59% | 80.06% | 91.32% |
| | 97.80% | 72.03% | 84.57% | 95.50% |
| | 97.50% | 75.88% | 84.57% | 96.14% |
| Gender-Bias | 99.30% | 55.57% | 70.06% | 68.76% |
| | 99.00% | 80.00% | 73.91% | 78.91% |
| | 98.70% | 84.91% | 85.00% | 88.44% |
| | 98.40% | 79.61% | 88.76% | 90.10% |
| | 98.10% | 81.76% | 90.47% | 90.73% |
| | 97.80% | 89.54% | 90.88% | 91.17% |
| | 97.50% | 90.05% | 90.95% | 91.05% |
| Country-Capital | 99.30% | -0.37% | 52.89% | 65.36% |
| | 99.00% | 13.22% | 75.42% | 82.50% |
| | 98.70% | 21.23% | 82.31% | 90.32% |
| | 98.40% | 30.35% | 86.41% | 91.62% |
| | 98.10% | 35.57% | 90.50% | 90.32% |
| | 97.80% | 57.54% | 91.43% | 91.81% |
| | 97.50% | 60.52% | 91.62% | 93.67% |
| Hypernymy | 99.30% | 7.41% | 10.11% | 14.01% |
| | 99.00% | 10.30% | 17.26% | 24.32% |
| | 98.70% | 17.11% | 29.28% | 38.27% |
| | 98.40% | 29.57% | 38.57% | 46.22% |
| | 98.10% | 35.51% | 47.27% | 54.93% |
| | 97.80% | 40.31% | 53.07% | 59.48% |
| | 97.50% | 45.58% | 55.62% | 67.24% |
| Greater-Than | 99.30% | 70.24% | 81.89% | 85.83% |
| | 99.00% | 92.52% | 95.35% | 94.88% |
| | 98.70% | 94.41% | 98.43% | 97.95% |
| | 98.40% | 95.43% | 98.03% | 98.98% |
| | 98.10% | 95.12% | 98.66% | 99.69% |
| | 97.80% | 95.98% | 100.00% | 99.53% |
| | 97.50% | 96.38% | 99.45% | 98.98% |

Table 5: Normalized faithfulness for Six Tasks in GPT2-Medium.

| Task | Edge Sparsity (%) | EAP (%) | EAP-IG (%) | EAP-GP (%) |
|---|---|---|---|---|
| | 98.38% | 46.71% | 60.03% | 64.75% |
| | 97.98% | 57.24% | 62.95% | 70.04% |
| | 97.58% | 62.44% | 69.54% | 72.94% |
| | 97.19% | 65.75% | 75.52% | 76.83% |
| IOI | 96.79% | 70.70% | 77.67% | 78.03% |
| | 99.17% | 82.11% | 91.05% | 93.37% |
| | 98.77% | 89.52% | 94.33% | 98.19% |
| | 98.38% | 94.56% | 96.08% | 98.81% |
| | 97.98% | 95.40% | 97.09% | 99.09% |
| | 97.58% | 96.16% | 98.82% | 98.40% |
| SVA | 97.19% | 96.26% | 99.00% | 98.73% |
| | 96.79% | 96.62% | 99.14% | 98.89% |
| | 99.17% | 77.20% | 80.17% | 83.28% |
| | 98.77% | 77.63% | 81.86% | 85.46% |
| | 98.38% | 81.46% | 86.47% | 88.69% |
| | 97.98% | 85.61% | 90.92% | 90.74% |
| | 97.58% | 88.74% | 91.42% | 92.85% |
| Gender-Bias | 97.19% | 90.87% | 94.11% | 95.52% |
| | 96.79% | 94.62% | 98.30% | 97.82% |
| | 99.17% | 42.73% | 71.34% | 75.24% |
| | 98.77% | 54.78% | 75.72% | 78.88% |
| | 98.38% | 62.91% | 78.74% | 84.78% |
| | 97.98% | 71.47% | 81.36% | 85.25% |
| | 97.58% | 75.52% | 84.19% | 87.35% |
| Country-Capital | 97.19% | 79.23% | 85.58% | 87.76% |
| | 96.79% | 80.45% | 86.16% | 88.16% |
| | 99.17% | 33.42% | 42.16% | 54.82% |
| | 98.77% | 41.45% | 52.54% | 64.08% |
| | 98.38% | 51.24% | 59.53% | 72.10% |
| | 97.98% | 56.07% | 65.81% | 76.61% |
| | 97.58% | 61.14% | 69.87% | 79.55% |
| Hypernymy | 97.19% | 64.88% | 74.83% | 82.08% |
| | 96.79% | 68.53% | 77.98% | 87.02% |
| | 99.17% | 79.90% | 83.23% | 87.96% |
| | 98.77% | 82.99% | 87.50% | 91.78% |
| | 98.38% | 84.00% | 89.47% | 94.71% |
| | 97.98% | 89.84% | 91.60% | 95.15% |
| | 97.58% | 91.63% | 93.96% | 97.90% |
| Greater-Than | 97.19% | 97.64% | 92.85% | 99.19% |
| | 96.79% | 96.68% | 95.76% | 99.16% |

Table 6: Normalized faithfulness for Six Tasks in GPT2-XL.

| Task | Edge Sparsity (%) | EAP (%) | EAP-IG (%) | EAP-GP (%) |
|------|-------------------|---------|------------|------------|
| | 99.19% | 55.63% | 60.02% | 61.43% |
| | 98.79% | 64.71% | 69.69% | 68.59% |
| | 98.39% | 73.07% | 74.77% | 74.51% |
| | 97.99% | 78.55% | 77.99% | 80.24% |
| | 97.59% | 82.27% | 81.33% | 82.97% |
| IOI | 97.19% | 83.91% | 84.90% | 86.08% |
| | 96.79% | 85.20% | 85.52% | 88.14% |
| | 99.20% | 90.17% | 93.69% | 96.22% |
| | 98.80% | 94.28% | 96.27% | 98.00% |
| | 98.40% | 96.30% | 97.73% | 98.30% |
| | 98.00% | 97.10% | 98.66% | 98.25% |
| | 97.60% | 97.53% | 98.97% | 98.57% |
| SVA | 97.20% | 98.01% | 99.18% | 98.75% |
| | 96.80% | 98.38% | 99.39% | 99.06% |
| | 98.80% | 98.26% | 98.96% | 99.58% |
| | 98.40% | 98.85% | 99.36% | 99.91% |
| | 98.00% | 99.25% | 99.63% | 100% |
| | 97.60% | 99.67% | 100% | 100% |
| | 97.20% | 99.67% | 100% | 100% |
| Gender-Bias | 96.80% | 99.58% | 100% | 100% |
| | 99.20% | 43.96% | 74.92% | 83.46% |
| | 98.80% | 64.10% | 81.29% | 86.78% |
| | 98.40% | 72.07% | 85.23% | 89.97% |
| | 98.00% | 78.09% | 88.18% | 91.28% |
| | 97.60% | 80.66% | 89.28% | 92.42% |
| Country-Capital | 97.20% | 85.00% | 90.91% | 93.49% |
| | 96.80% | 87.06% | 92.03% | 94.17% |
| | 99.20% | 55.16% | 60.54% | 74.73% |
| | 98.80% | 68.55% | 73.93% | 84.02% |
| | 98.40% | 76.64% | 82.15% | 88.42% |
| | 98.00% | 82.60% | 86.41% | 91.24% |
| | 97.60% | 86.50% | 88.67% | 93.61% |
| Hypernymy | 97.20% | 88.79% | 90.94% | 95.23% |
| | 96.80% | 90.71% | 92.79% | 95.92% |
| | 98.80% | 97.96% | 98.15% | 98.68% |
| | 98.40% | 98.30% | 98.75% | 98.91% |
| | 98.00% | 98.15% | 98.74% | 99.08% |
| | 97.60% | 98.47% | 98.61% | 99.11% |
| | 97.20% | 98.53% | 98.75% | 99.13% |
| Greater-Than | 96.80% | 99.31% | 98.89% | 99.45% |

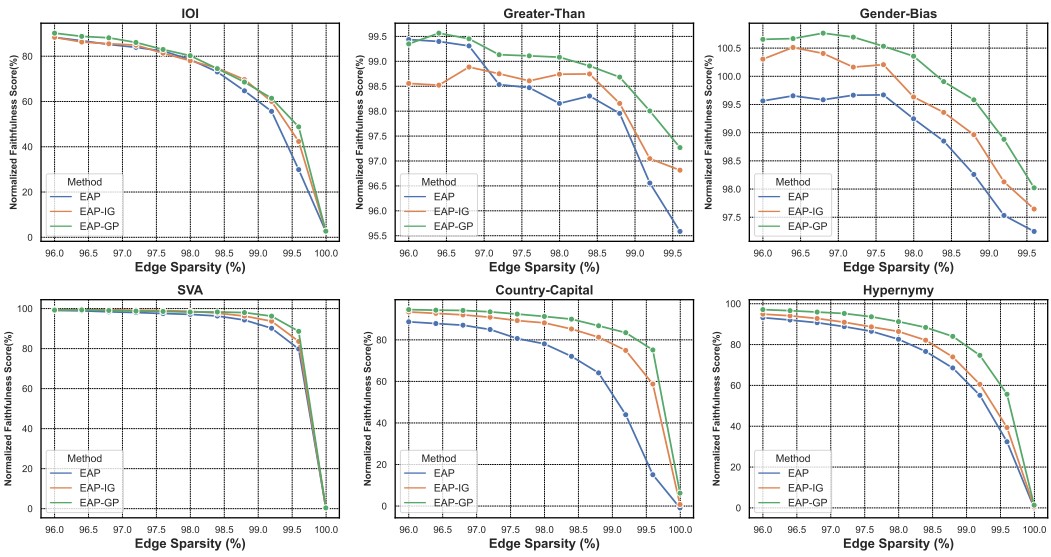

Figure 6: Comparison of circuit performance across different methods on GPT-2 XL. In all plots, a higher value indicates better performance. EAP-GP identifies circuits that outperform other methods across all six tasks.

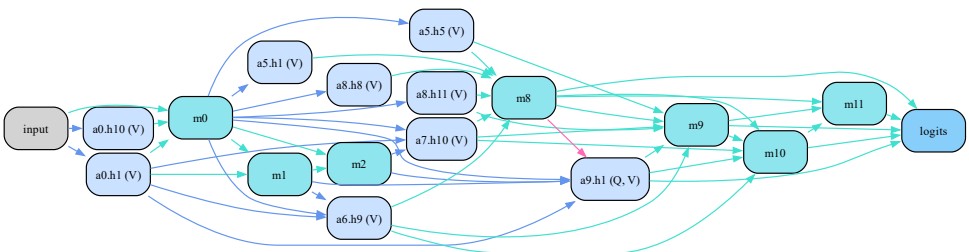

Figure 7: A circuit for Greater-Than with 99.9% sparsity, found by EAP-GP.

