# OpenReview forum: "EAP-GP: Mitigating Saturation Effect in Gradient-based Automated Circuit Identification"
_NeurIPS.cc/2025/Conference — NeurIPS 2025 poster_

### Official Review · Reviewer_LBjM · 2025-07-02

**Clarity:** 3
**Significance:** 2
**Originality:** 3
**Rating:** 5
**Confidence:** 4

**Summary:**

This work focuses on improving gradient-based circuit discovery techniques that aim to reverse engineer neural networks by analyzing their computational graph. In particular, the authors propose Edge Attribution Patching with GradPath (EAP-GP), which improves over existing techniques such as EAP and EAP-IG by avoiding zones of gradient saturation. The paper reports improvements of up to 17% in circuit faithfulness.

**Questions:**

Improve motivation of the work: What is the theory of impact for this work? What is the utter goal of having more faithful circuits?

Can we make EAP-GP faster so that it's more competitive against EAP-IG?

Are there any limitations on using manually found ground-truth circuits as a part of a quantitative experiment?

Does it make sense to do a qualitative analysis of the circuits found by these techniques?

**Ethical Concerns:**

["NO or VERY MINOR ethics concerns only"]

**Final Justification:**

The authors have compellingly addressed my concerns on the ground-truth circuits used for comparison. They will also improve the final work by further discussing limitations and future work (such as making EAP-GP cheaper to run), and by including qualitative analysis of some of the circuits found (which they already did for the rebuttal). I remain positive about this paper and believe it is a technically solid advance on the field of Mechanistic Interpretability.

**Limitations:**

Authors have addressed some limitations in the Experiments section, but I think the paper would benefit from having a dedicated section. As mentioned above in the review, authors should also make explicit that EAP-GP is way more costly than EAP-IG (i.e., there's a tradeoff compute/accuracy between techniques).

**Paper Formatting Concerns:**

No major formatting issues.

**Quality:**

3

**Strengths And Weaknesses:**

Strengths:
- The work seems very relevant for the field of Mechanistic Interpretability: Circuit Discovery techniques are constrained by the quality and cost of the metrics for measuring the importance of each edge/node in the neural network computational graph.
- EAP-GP seems to be a sound proposal that improves over the state of the art (EAP and EAP-IG)
- The work includes a solid experimental setup measuring EAP-GP vs EAP and EAP-IG on 6 different tasks and 3 different variants of GPT-2 (small, medium, and XL)
- The authors also compare the computation time for EAP-GP vs EAP and EAP-IG, which is important to understand the tradeoffs between these techniques
- Writing is very clear and concise for the most part

Weaknesses:
- No mention of limitations or future work. Are there any edge cases that might be problematic for EAP-GP? Can we make EAP-GP cheaper to run (since it seems to be ~4x more expensive than EAP-IG)? Etc.
- Unclear if the ground-truth circuits used for comparison (Syed et al) provide a good signal or not on the quality of the circuits found automatically. I.e., if these ground-truth circuits are not very faithful, then EAP-GP having less precision/recall than EAP-IG might be good.
- A qualitative analysis of some of the circuits found (e.g., for IOI) would be nice. Does EAP-GP find attention heads or MLPs that are interesting or relevant for the circuit and haven't been analysed before?
- Unclear what the practical applications of having more faithful circuits for AI Safety are. Also, circuit discovery might not scale to bigger SOTA models.

Minor issues and typos:
- The main body should reference the compute analysis in Table 3, which seems to suggest that EAP-GP is ~4x more expensive to run than EAP-IG.
- Maybe rename Section 6 to "Evaluation" instead of "Experiments"
- Missing an explicit mention of the AUCs for all methods in the precision-recall curves of Fig. 4.
- Paper goes back and forth with these terms: baseline input, clean input, corrupted input, original input, . Please choose two and go with them, otherwise is quite confusing for the reader.
- Faithfulness of circuits is talked about in intro (line 51), way before it's defined in the paper, which is slightly confusing if you don't know before hand what it means.
- Typo in line 125, should be "graph" instead of "subgraph". I.e., "Given a model G, which can be represented as a computational **graph**"
- IOI dataset is mentioned for the first time in line 165, without any cite. Please add the cite or reference to the appropriate section where this is explained.
- Typo in line 211: "iteratively" twice in the same sentence.

---

> ### Author Rebuttal · Authors · 2025-07-31
>
> >W1: No mention of limitations or future work. Are there any edge cases that might be problematic for EAP-GP? Can we make EAP-GP cheaper to run (since it seems to be ~4x more expensive than EAP-IG)? Etc.
>
>
> A1: Thank you for the thoughtful comments. We acknowledge that EAP-GP introduces additional computational cost (∼4× slower than EAP-IG, as shown in Table 3), primarily due to the dynamic construction of model-dependent integration paths. In future work, we plan to explore efficiency improvements such as low-rank approximations, early stopping based on gradient convergence, and path simplification techniques.
>
> As for limitations, EAP-GP may underperform in cases where gradients near corrupted inputs are highly unstable or noisy—especially in tasks with weak supervision signals. We view this as an opportunity for further robustness improvements, such as incorporating gradient smoothing or hybrid attribution strategies. We will add a dedicated section at the end of the paper to discuss these limitations and future directions.
>
>
> >W2: Unclear if the ground-truth circuits used for comparison (Syed et al) provide a good signal or not on the quality of the circuits found automatically. I.e., if these ground-truth circuits are not very faithful, then EAP-GP having less precision/recall than EAP-IG might be good.
>
>
> A2: Thank you for the insightful suggestion. In the context of circuit discovery, the primary goal of automatic methods is to approximate both the process and the quality of manual circuit discovery. Since both approaches rely on the same evaluation metrics (e.g., logit difference) to identify circuits, it is natural to validate their consistency through qualitative analysis. By qualitatively comparing the circuits discovered by automatic methods with those obtained manually, we can assess whether they capture the same interpretable mechanisms. This offers further evidence that automatic circuit discovery can faithfully reproduce the insights of manual analysis and scale to more complex settings, making it a practical and effective alternative.
>
>
> >W3: A qualitative analysis of some of the circuits found (e.g., for IOI) would be nice. Does EAP-GP find attention heads or MLPs that are interesting or relevant for the circuit and haven't been analysed before?
>
>
> A3:Thank you for the insightful suggestion. We conducted a qualitative comparison of the circuits discovered by EAP, EAP-IG, and EAP-GP on the IOI task. Below are representative circuits extracted by each method.
>
> **EAP:** input-&gt;a0.h1&lt;v&gt;,input-&gt;m0,a0.h1-&gt;m0,a0.h1-&gt;m1,m0-&gt;m1,a0.h1-&gt;m2,m0-&gt;m2,m1-&gt;m2,m0-&gt;m3,m2-&gt;m3,m1-&gt;a6.h9&lt;v&gt;,a0.h1-&gt;a7.h10&lt;v&gt;,m0-&gt;a7.h10&lt;v&gt;,m2-&gt;a7.h10&lt;v&gt;,m0-&gt;a8.h8&lt;v&gt;,m0-&gt;a8.h11&lt;v&gt;,a6.h9-&gt;m8,a7.h10-&gt;m8,a8.h8-&gt;m8,a8.h11-&gt;m8,a0.h1-&gt;a9.h1&lt;v&gt;,m0-&gt;a9.h1&lt;v&gt;,m1-&gt;a9.h1&lt;v&gt;,m2-&gt;a9.h1&lt;v&gt;,m3-&gt;a9.h1&lt;v&gt;,a7.h10-&gt;m9,a8.h11-&gt;m9,m8-&gt;m9,a9.h1-&gt;m9,m0-&gt;a10.h7&lt;v&gt;,a7.h10-&gt;m10,a8.h11-&gt;m10,m8-&gt;m10,a9.h1-&gt;m10,m9-&gt;m10,m8-&gt;m11,m9-&gt;m11,m10-&gt;m11,m8-&gt;logits,a9.h1-&gt;logits,m9-&gt;logits,a10.h7-&gt;logits,m10-&gt;logits,m11-&gt;logits
>
>
> **EAP-IG:** input-&gt;a0.h1&lt;v&gt;,input-&gt;a0.h10&lt;v&gt;,input-&gt;m0,a0.h1-&gt;m0,a0.h10-&gt;m0,m0-&gt;m1,m0-&gt;m2,m1-&gt;m2,m0-&gt;m3,m1-&gt;m3,m0-&gt;a5.h1&lt;v&gt;,m0-&gt;a5.h5&lt;v&gt;,a0.h1-&gt;a6.h9&lt;v&gt;,m0-&gt;a6.h9&lt;v&gt;,m1-&gt;a6.h9&lt;v&gt;,a0.h1-&gt;a7.h10&lt;v&gt;,m0-&gt;a7.h10&lt;v&gt;,m2-&gt;a7.h10&lt;v&gt;,m0-&gt;a8.h8&lt;v&gt;,m0-&gt;a8.h11&lt;v&gt;,a5.h1-&gt;m8,a6.h9-&gt;m8,a7.h10-&gt;m8,a8.h8-&gt;m8,a8.h11-&gt;m8,a0.h1-&gt;a9.h1&lt;v&gt;,m0-&gt;a9.h1&lt;v&gt;,m1-&gt;a9.h1&lt;v&gt;,m2-&gt;a9.h1&lt;v&gt;,m3-&gt;a9.h1&lt;v&gt;,m8-&gt;a9.h1&lt;q&gt;,a5.h5-&gt;m9,a6.h9-&gt;m9,a7.h10-&gt;m9,a8.h11-&gt;m9,m8-&gt;m9,a9.h1-&gt;m9,a6.h9-&gt;m10,a7.h10-&gt;m10,m8-&gt;m10,a9.h1-&gt;m10,m9-&gt;m10,m8-&gt;m11,m9-&gt;m11,m10-&gt;m11,m8-&gt;logits,a9.h1-&gt;logits,m9-&gt;logits,m10-&gt;logits,m11-&gt;logits
>
>
> **EAP-GP:** input-&gt;a0.h1&lt;v&gt;,input-&gt;a0.h10&lt;v&gt;,input-&gt;m0,a0.h1-&gt;m0,a0.h10-&gt;m0,a0.h1-&gt;m1,m0-&gt;m1,m0-&gt;m2,m1-&gt;m2,m0-&gt;a5.h1&lt;v&gt;,m0-&gt;a5.h5&lt;v&gt;,a0.h1-&gt;a6.h9&lt;v&gt;,m0-&gt;a6.h9&lt;v&gt;,m1-&gt;a6.h9&lt;v&gt;,a0.h1-&gt;a7.h10&lt;v&gt;,m0-&gt;a7.h10&lt;v&gt;,m2-&gt;a7.h10&lt;v&gt;,m0-&gt;a8.h8&lt;v&gt;,m0-&gt;a8.h11&lt;v&gt;,a5.h1-&gt;m8,a5.h5-&gt;m8,a6.h9-&gt;m8,a7.h10-&gt;m8,a8.h8-&gt;m8,a8.h11-&gt;m8,a0.h1-&gt;a9.h1&lt;v&gt;,m0-&gt;a9.h1&lt;v&gt;,m1-&gt;a9.h1&lt;v&gt;,m2-&gt;a9.h1&lt;v&gt;,m8-&gt;a9.h1&lt;q&gt;,a5.h5-&gt;m9,a6.h9-&gt;m9,a7.h10-&gt;m9,a8.h11-&gt;m9,m8-&gt;m9,a9.h1-&gt;m9,a6.h9-&gt;m10,a7.h10-&gt;m10,m8-&gt;m10,a9.h1-&gt;m10,m9-&gt;m10,m8-&gt;m11,m9-&gt;m11,m10-&gt;m11,m8-&gt;logits,a9.h1-&gt;logits,m9-&gt;logits,m10-&gt;logits,m11-&gt;logits
>
> The circuits discovered by EAP, EAP-IG, and EAP-GP on the IOI task demonstrate progressively more advanced patterns of information flow and integration. EAP identifies a sparse, mostly linear circuit centered on a single attention head, resulting in limited feature fusion and low redundancy. EAP-IG builds upon this by introducing multiple attention heads and localized modular interactions, enabling richer internal connectivity and the early use of query-based aggregation. EAP-GP further advances these capabilities by leveraging its gradient-guided path integration design to identify attribution paths that avoid saturated or misleading gradients. As a result, it discovers circuits involving a broader set of attention heads across multiple layers, centralizes information through a key convergence node (e.g., m8), and forms deep, multi-branch structures with extensive cross-layer integration. This reflects EAP-GP's ability to uncover more faithful and causally grounded circuits. As a result, the discovered structures exhibit significantly higher structural complexity, redundancy, and robustness. These properties enable the final logits to be influenced by diverse and complementary sources of information.
>
>
>
> >W4: Unclear what the practical applications of having more faithful circuits for AI Safety are. Also, circuit discovery might not scale to bigger SOTA models.
>
> A4:Thank you for raising these concerns. We believe that improving circuit faithfulness has important implications for AI safety. Faithful circuits allow researchers to uncover the true causal mechanisms behind model behavior, which is essential for tasks such as auditing, detecting spurious correlations, and ensuring behavioral alignment in high-stakes applications (e.g., content moderation, planning, or question answering).
>
> Importantly, much of mechanistic interpretability is grounded in attribution analysis, and our work reveals saturation as a previously underappreciated failure mode of gradient-based attribution in circuit discovery. This issue may also affect other attribution-based tools widely used in AI safety and interpretability, highlighting the need for methods that produce faithful and robust causal explanations. Without such faithfulness, interventions or explanations may be misleading or ineffective.
>
> Regarding scalability, earlier circuit discovery approaches relied heavily on manual inspection. Recent advances, including our proposed EAP-GP, demonstrate that gradient-based methods are capable of scaling to larger models. We acknowledge that challenges remain for frontier-scale systems, but view our work as a meaningful step toward scalable and reliable mechanistic interpretability.
>
>
> > Minor issues and typos:
>
> A5: Thank you for your careful reading and constructive suggestions. We will revise the paper accordingly. Specifically, we will reference the compute analysis in Table 3 to highlight the ~4× cost of EAP-GP, rename Section 6 to “Evaluation,” include AUC values in Fig. 4, and standardize terminology to “clean input” and “corrupted input” for clarity. We will also clarify the definition of circuit faithfulness earlier in the text, fix the typo in line 125 (“graph”), add a citation or reference for the IOI dataset at its first mention, and remove the duplicated “iteratively” in line 211. Your feedback is much appreciated and will help improve the overall clarity and consistency of the paper.
>
>
>
>
>
> >Q1: Improve motivation of the work: What is the theory of impact for this work? What is the utter goal of having more faithful circuits?
>
> A6: Thank you for the Question. Please refer to our response to W4 for details.
>
> >Q2:Can we make EAP-GP faster so that it's more competitive against EAP-IG?
>
> A7: Thank you for the Question. Please refer to our response to W1 for details.
>
> >Q3: Are there any limitations on using manually found ground-truth circuits as a part of a quantitative experiment?
>
> A8: Thank you for the Question. Please refer to our response to W2 for details.
>
> >Q4:Does it make sense to do a qualitative analysis of the circuits found by these techniques?
>
> A9: Thank you for the Question. Please refer to our response to W3 for details.

---

> > ### Comment · Reviewer_LBjM · 2025-08-04
> >
> > Thank you for your detailed response and willingness to improve the work. I will keep my score.

---

### Official Review · Reviewer_vD2V · 2025-07-03

**Clarity:** 3
**Significance:** 3
**Originality:** 3
**Rating:** 5
**Confidence:** 3

**Summary:**

This paper presents an improvement to the gradient-based automatic circuit discovery method EAP and its follow-up, EAP-IG. These methods identify limitations in prior gradient-based approaches to circuit discovery. In particular, such methods compute an edge's importance by multiplying the gradient with the change in activation, and, therefore, EAP suffers from the zero-gradient problem: when the gradient approaches zero, the importance score also approaches zero, regardless of the actual significance of the edge. EAP-IG was proposed to mitigate this issue by using integrated gradients and achieved some improvements. However, it remains affected by gradient saturation: if the corrupted input enters a "saturation zone," the gradient again becomes nearly zero.

The authors proposed GradPath to mitigate this issue. EAP-IG can be seen as applying a "big jump" along a straight line, which may lead the method into a saturation region, resulting in a small gradient. Instead of integrating in one step, the authors' method iteratively follows the steepest steps one at a time, producing a "GradPath" that mitigates the saturation effect. As a result, EAP-GP achieves better performance than both EAP and EAP-IG.

**Questions:**

- What do you think the better performance of EAP-GP can bring for the interpretability community?

- Can any misconceptions be refuted, or previously overlooked points revealed, based on the improved performance? You do not need to do new experiments, but please pointing out some possible directions.

**Ethical Concerns:**

["NO or VERY MINOR ethics concerns only"]

**Final Justification:**

Thanks for the response. I am satisfied with the elaboration. Please incorporate them in the updated draft. I will maintain my positive assessment.

**Limitations:**

yes

**Quality:**

3

**Strengths And Weaknesses:**

Overall, this paper presents a novel and effective improvement to gradient-based automatic circuit discovery methods, offering a useful mitigation strategy for the saturation effect. The main limitation is that it does not provide any new insights into the interpretation of LMs, which should be the primary goal of circuit discovery. This is concerning, as such work risks turning automatic circuit discovery into a leaderboard benchmarking task, drifting away from its original purpose: understanding the internal workings of LLMs. Nonetheless, this work provides a new tool to the community that shows promise as a generalisable approach. I believe it can support further research and help others develop more effective interpretations in the future.

Strength
- This paper is very well written. The presentation of the paper is clear and easy to follow.
- The authors identify that the saturation effect for EAP-IG, which suggests that a less severe version of the zero gradient problem also exists for EAP-IG.
- This work presents solid performance-based evaluation and showcased the state-of-the-art performance.
Weakness
- Although this work concerns interpretability, the paper does not offer new insights into the interpretation of the LM. This is acceptable for the this work, but it would still be beneficial to include at least a discussion of the interpretability aspect. The paper would be significantly stronger if the authors could use EAP-GP's new, improved result to refute previous misconceptions or reveal previously overlooked points caused by the limitation of EAP or EAP-IG.

---

> ### Author Rebuttal · Authors · 2025-07-31
>
> >Q1: What do you think the better performance of EAP-GP can bring for the interpretability community?
>
> A1: Thank you for the constructive question. We believe the improved performance of EAP-GP has several important implications for the interpretability community:
>
> **More Reliable Circuits Under Sparsity Constraints:** EAP-GP consistently discovers more faithful circuits under high sparsity levels (see Table 4), particularly in complex reasoning tasks. This enables researchers to analyze much smaller subgraphs without sacrificing interpretability fidelity, which is critical for scaling interpretability to larger models.
>
> **Recognizing Saturation as a General Challenge:** By identifying and addressing saturation effects, EAP-GP highlights a previously under-discussed failure mode in mechanistic interpretability methods. While saturation has been noted in other contexts, to our knowledge, this is the first work to explicitly observe and address it within circuit discovery. This finding may prompt a broader re-examination of gradient-based attribution tools used throughout mechanistic interpretability.
>
> **Towards Path-Adaptive Attribution Methods**    EAP-GP offers a new perspective on attribution by introducing a general design principle: adapting the integration path to the model’s local gradient landscape. This perspective encourages future research into dynamic, model-aware attribution strategies that are better equipped to handle common issues such as saturation and vanishing gradients.
>
>
> > Q2: Can any misconceptions be refuted, or previously overlooked points revealed, based on the improved performance? You do not need to do new experiments, but please pointing out some possible directions.
>
>
> A4: Thank you for the insightful suggestion. We conducted a qualitative comparison of the circuits discovered by EAP, EAP-IG, and EAP-GP on the IOI task. Below are representative circuits extracted by each method.
>
> **EAP:** input-&gt;a0.h1&lt;v&gt;,input-&gt;m0,a0.h1-&gt;m0,a0.h1-&gt;m1,m0-&gt;m1,a0.h1-&gt;m2,m0-&gt;m2,m1-&gt;m2,m0-&gt;m3,m2-&gt;m3,m1-&gt;a6.h9&lt;v&gt;,a0.h1-&gt;a7.h10&lt;v&gt;,m0-&gt;a7.h10&lt;v&gt;,m2-&gt;a7.h10&lt;v&gt;,m0-&gt;a8.h8&lt;v&gt;,m0-&gt;a8.h11&lt;v&gt;,a6.h9-&gt;m8,a7.h10-&gt;m8,a8.h8-&gt;m8,a8.h11-&gt;m8,a0.h1-&gt;a9.h1&lt;v&gt;,m0-&gt;a9.h1&lt;v&gt;,m1-&gt;a9.h1&lt;v&gt;,m2-&gt;a9.h1&lt;v&gt;,m3-&gt;a9.h1&lt;v&gt;,a7.h10-&gt;m9,a8.h11-&gt;m9,m8-&gt;m9,a9.h1-&gt;m9,m0-&gt;a10.h7&lt;v&gt;,a7.h10-&gt;m10,a8.h11-&gt;m10,m8-&gt;m10,a9.h1-&gt;m10,m9-&gt;m10,m8-&gt;m11,m9-&gt;m11,m10-&gt;m11,m8-&gt;logits,a9.h1-&gt;logits,m9-&gt;logits,a10.h7-&gt;logits,m10-&gt;logits,m11-&gt;logits
>
>
> **EAP-IG:** input-&gt;a0.h1&lt;v&gt;,input-&gt;a0.h10&lt;v&gt;,input-&gt;m0,a0.h1-&gt;m0,a0.h10-&gt;m0,m0-&gt;m1,m0-&gt;m2,m1-&gt;m2,m0-&gt;m3,m1-&gt;m3,m0-&gt;a5.h1&lt;v&gt;,m0-&gt;a5.h5&lt;v&gt;,a0.h1-&gt;a6.h9&lt;v&gt;,m0-&gt;a6.h9&lt;v&gt;,m1-&gt;a6.h9&lt;v&gt;,a0.h1-&gt;a7.h10&lt;v&gt;,m0-&gt;a7.h10&lt;v&gt;,m2-&gt;a7.h10&lt;v&gt;,m0-&gt;a8.h8&lt;v&gt;,m0-&gt;a8.h11&lt;v&gt;,a5.h1-&gt;m8,a6.h9-&gt;m8,a7.h10-&gt;m8,a8.h8-&gt;m8,a8.h11-&gt;m8,a0.h1-&gt;a9.h1&lt;v&gt;,m0-&gt;a9.h1&lt;v&gt;,m1-&gt;a9.h1&lt;v&gt;,m2-&gt;a9.h1&lt;v&gt;,m3-&gt;a9.h1&lt;v&gt;,m8-&gt;a9.h1&lt;q&gt;,a5.h5-&gt;m9,a6.h9-&gt;m9,a7.h10-&gt;m9,a8.h11-&gt;m9,m8-&gt;m9,a9.h1-&gt;m9,a6.h9-&gt;m10,a7.h10-&gt;m10,m8-&gt;m10,a9.h1-&gt;m10,m9-&gt;m10,m8-&gt;m11,m9-&gt;m11,m10-&gt;m11,m8-&gt;logits,a9.h1-&gt;logits,m9-&gt;logits,m10-&gt;logits,m11-&gt;logits
>
>
> **EAP-GP:** input-&gt;a0.h1&lt;v&gt;,input-&gt;a0.h10&lt;v&gt;,input-&gt;m0,a0.h1-&gt;m0,a0.h10-&gt;m0,a0.h1-&gt;m1,m0-&gt;m1,m0-&gt;m2,m1-&gt;m2,m0-&gt;a5.h1&lt;v&gt;,m0-&gt;a5.h5&lt;v&gt;,a0.h1-&gt;a6.h9&lt;v&gt;,m0-&gt;a6.h9&lt;v&gt;,m1-&gt;a6.h9&lt;v&gt;,a0.h1-&gt;a7.h10&lt;v&gt;,m0-&gt;a7.h10&lt;v&gt;,m2-&gt;a7.h10&lt;v&gt;,m0-&gt;a8.h8&lt;v&gt;,m0-&gt;a8.h11&lt;v&gt;,a5.h1-&gt;m8,a5.h5-&gt;m8,a6.h9-&gt;m8,a7.h10-&gt;m8,a8.h8-&gt;m8,a8.h11-&gt;m8,a0.h1-&gt;a9.h1&lt;v&gt;,m0-&gt;a9.h1&lt;v&gt;,m1-&gt;a9.h1&lt;v&gt;,m2-&gt;a9.h1&lt;v&gt;,m8-&gt;a9.h1&lt;q&gt;,a5.h5-&gt;m9,a6.h9-&gt;m9,a7.h10-&gt;m9,a8.h11-&gt;m9,m8-&gt;m9,a9.h1-&gt;m9,a6.h9-&gt;m10,a7.h10-&gt;m10,m8-&gt;m10,a9.h1-&gt;m10,m9-&gt;m10,m8-&gt;m11,m9-&gt;m11,m10-&gt;m11,m8-&gt;logits,a9.h1-&gt;logits,m9-&gt;logits,m10-&gt;logits,m11-&gt;logits
>
>
> The circuits discovered by EAP, EAP-IG, and EAP-GP on the IOI task demonstrate progressively more advanced patterns of information flow and integration. EAP identifies a sparse, mostly linear circuit centered on a single attention head, resulting in limited feature fusion and low redundancy. EAP-IG builds upon this by introducing multiple attention heads and localized modular interactions, enabling richer internal connectivity and the early use of query-based aggregation. EAP-GP further advances these capabilities by leveraging its gradient-guided path integration design to identify attribution paths that avoid saturated or misleading gradients. As a result, it discovers circuits involving a broader set of attention heads across multiple layers, centralizes information through a key convergence node (e.g., m8), and forms deep, multi-branch structures with extensive cross-layer integration. This reflects EAP-GP's ability to uncover more faithful and causally grounded circuits. As a result, the discovered structures exhibit significantly higher structural complexity, redundancy, and robustness. These properties enable the final logits to be influenced by diverse and complementary sources of information.

---

### Official Review · Reviewer_ptDW · 2025-07-03

**Clarity:** 2
**Significance:** 2
**Originality:** 3
**Rating:** 5
**Confidence:** 4

**Summary:**

This work introduces Edge Attribution Patching with GradPath (EAP-GP), an improvement to an existing circuit discovery algorithm, EAP-IG. This work identifies a problem with EAP-IG (i.e., that integrated gradients pushes the model along a path that may traverse through a region of near-zero gradient) which hinders its attributions to connections between model components, and then introduces a simple solution to that problem. Across several evaluations, EAP-GP outperforms existing methods in both faithfulness and consistency with manually annotated circuits.

**Questions:**

Why does EAP-GP not still produce performance gains at very high sparsities if these tasks are basically solved by simple feature extraction?

How ubiquitous is the saturation effect within a model, across models, across tasks?

**Ethical Concerns:**

["NO or VERY MINOR ethics concerns only"]

**Final Justification:**

The authors promised the most important revisions/clarifications that I requested during the initial review, and have conducted additional analyses to address my concern regarding the ubiquity of saturation effects.

**Limitations:**

Limitations are discussed at the end of the results section, but should be put in a limitations section, with a disclaimer that the method is sensitive to hyperparameters

**Quality:**

3

**Strengths And Weaknesses:**

The problem that authors identify with EAP-IG is reasonable, as is their solution. Though the method empirically outperforms across a variety of tasks, I am left wondering how ubiquitous the saturation effects are. Section 4 plots only one edge in the IOI task, and could be improved by including similar plots aggregated over all tasks, or more comprehensive discussions of e.g., how many edges face these saturation effects.

Additionally, further discussion of the greater-than and gender-bias results is warranted. Why does EAP-GP not still produce performance gains at very high sparsities if these tasks are basically solved by simple feature extraction?

Writing:
The "zero-gradients problem" is introduced early on, but not explicitly defined at that point. This will confuse some readers, so please define.

Finally, limitations are discussed at the end of the results section, but should be put in a limitations section, with a disclaimer that the method is sensitive to hyperparameters

---

> ### Author Rebuttal · Authors · 2025-07-31
>
> >W1: how ubiquitous the saturation effects are?
>
> A1: Thank you for the Question. Please refer to our response to Q2 for details.
>
> >W2: further discussion of the greater-than and gender-bias results is warranted
>
> A2: Thank you for the Question. Please refer to our response to Q1 for details.
>
> >W3: Writing: The "zero-gradients problem" is introduced early on, but not explicitly defined at that point. This will confuse some readers, so please define.
>
> A3: Thank you for the suggestion. We will revise the manuscript to include an explicit definition of the “zero-gradients problem” when it is first introduced, to avoid potential confusion for readers.
>
> >W4: limitations are discussed at the end of the results section, but should be put in a limitations section, with a disclaimer that the method is sensitive to hyperparameters
>
> A4: Thank you for the suggestion. Please refer to our response to L1 for details.
>
>
>
>
> >Q1: Why does EAP-GP not still produce performance gains at very high sparsities if these tasks are basically solved by simple feature extraction?
>
> A5: Thank you for your thoughtful question. We agree that tasks like Greater-Than and Gender-Bias are largely solvable via simple feature extraction mechanisms. As shown in Figure 3 and Table 4 of our paper, the faithfulness scores of all three methods (EAP, EAP-IG, EAP-GP) are indeed relatively close under high sparsity regimes in these tasks. However, there are a few important considerations explaining why EAP-GP does not yield further substantial gains in such cases:
>
> **Ceiling Effect due to Task Simplicity:** These tasks involve relatively shallow reasoning and are dominated by localized features (e.g., profession-to-pronoun or numeric increments). Consequently, even a small number of edges are sufficient to recover most of the task-relevant signal. Thus, all methods tend to converge to similarly high faithfulness scores under high sparsity, leaving limited headroom for improvement.
>
> **Circuit Saturation vs. Attribution Saturation:** EAP-GP is designed to mitigate saturation effects in attribution scores, not to discover more complex circuits where none are needed. In tasks that do not require long-range or compositional reasoning, even naive edge selection strategies may yield circuits that already match the model's behavior well. Hence, while EAP-GP still helps ensure robustness, its advantage becomes less pronounced.
>
> **Sensitivity to Sparsity Thresholds:** At extremely high sparsity (e.g., >99.5%), all methods can become sensitive to small attribution variations, as shown in Figure 2. EAP-GP’s step size dynamics (see Section 6.2) may also amplify noise due to gradient instability near the baseline, which can slightly hurt performance compared to moderate sparsity.
>
> In summary, the marginal gains of EAP-GP in simple tasks under high sparsity are bounded not by its capability, but by the simplicity of the task structure itself. We believe this aligns with the broader finding that gradient saturation effects are more critical in complex tasks such as IOI and Hypernymy, where EAP-GP shows the most substantial improvements.
>
>
>
>
> >Q2: How ubiquitous is the saturation effect within a model, across models, across tasks?
>
> A6: We thank the reviewer for raising this important point regarding the generality of saturation effects. Our investigation reveals that the saturation effect is a widespread and recurring phenomenon, as detailed below:
>
> **Within a Model:** As shown in Figure 1 and Section 4, we observe saturation effects on edge gradients even within a single circuit in the IOI task. Specifically, the straight-line path used by EAP-IG frequently traverses regions where the gradient of the loss ∂L/∂x becomes nearly zero. These regions occur at both the start and end of the interpolation path, regardless of the specific edge being evaluated. This indicates that saturation is not limited to individual edges but affects broad regions of the activation space.
>
>
> | Step | Method  | Mean    | Method  | Mean   |
> |------|---------|---------|---------|--------|
> |   1  | EAP-GP  | 1.949   | EAP-IG  | 1.949  |
> |   2  | EAP-GP  | 1.294   | EAP-IG  | 0.019  |
> |   3  | EAP-GP  | 0.467   | EAP-IG  | 0.160  |
> |   4  | EAP-GP  | 0.225   | EAP-IG  | 0.076  |
> |   5  | EAP-GP  | 0.123   | EAP-IG  | 0.123  |
>
> To address this, we conducted additional experiments on the IOI task to measure the average gradient norm across all circuit edges during attribution. As shown in the supplementary table, EAP-IG tends to follow integration paths that pass through saturated regions, resulting in low gradient magnitudes and less accurate attributions. In contrast, our proposed EAP-GP avoids these saturated areas by adaptively selecting informative paths, leading to higher gradient norms and more faithful circuit discovery.
>
> **Across Tasks:** Figure 3 and Table 4 show that saturation effects are especially detrimental in complex reasoning tasks such as IOI, Hypernymy, and SVA. In these tasks, EAP-IG shows a noticeable drop in faithfulness compared to EAP-GP as sparsity increases, suggesting that saturation limits its ability to select faithful edges. In contrast, in simpler tasks like Greater-Than and Gender-Bias, where feature extraction is more direct, the saturation effect has less impact—but is still measurable (e.g., Table 4 shows 1–2% gains by EAP-GP at 99% sparsity).
>
> **Across Models:** As shown in Figures 5 and 6 (Appendix A.2), the saturation effect persists across GPT-2 Small, Medium, and XL. In particular, EAP-GP consistently outperforms EAP-IG across all model sizes, demonstrating that the issue is not alleviated by larger capacity or more parameters. In fact, larger models often exhibit more pronounced saturation due to their increased nonlinearity and overparameterization, further motivating the need for path-aware methods like EAP-GP.
>
> In conclusion, saturation is a general and systematic problem that arises across architectures, tasks, and layers. Our method mitigates this problem by dynamically adjusting the gradient path to avoid such regions, which proves effective and robust in a variety of experimental settings.
>
>
> >L1:Limitations are discussed at the end of the results section, but should be put in a limitations section, with a disclaimer that the method is sensitive to hyperparameters
>
> A7: Thank you for the suggestion. We agree that the paper would benefit from a clearer presentation of its limitations. To address this, we will move the current limitations discussion from the end of Section 6.2 into a new section titled Limitations and Future Work. This section will also mention that EAP-GP is somewhat sensitive to its main hyperparameter k, which controls the number of integration steps.

---

> > ### Comment · Reviewer_ptDW · 2025-08-04
> >
> > Thank you for the thoughtful and thorough response. I am convinced by the responses to my question regarding simple tasks and by the additional experiments addressing the generality of saturation effects. In response to these clarifications and analyses, I am raising my score to a 5.

---

> > > ### Author Response · Authors · 2025-08-04
> > >
> > > Thank you very much for your thoughtful reconsideration and for raising the score. We truly appreciate your engagement with our work and your constructive feedback throughout the review process. Your comments have helped us improve the clarity and depth of our analysis.

---

### Official Review · Reviewer_pkLe · 2025-07-14

**Clarity:** 4
**Significance:** 3
**Originality:** 3
**Rating:** 2
**Confidence:** 5

**Summary:**

This paper concerns methods for circuit discovery, that is, finding the minimal, complete and faithful subgraph of the NN's computational graph which is sufficient to perform the task. It identifies a shortcoming of the existing most popular method: EAP-IG. EAP-IG computes the gradient with respect to computational graph components, at various points in the straight line between activations at the clean input x and at corrupted input x', and uses the gradient's average as the "attribution score". Components with the highest attribution score are taken as the relevant subgraph, that is, the circuit.

The problem is that at many of these intermediate straight-line points, in LLMs, the gradient is saturated to zero. As a result, the attribution scores for many of the subgraph components is too low. The authors propose to instead take gradient descent steps from the clean towards the corrupted input, which has fewer saturation problems. As a result, they are able to improve the faithfulness scores ("loss recovered") by 10-17% (depending on the task)

**Questions:**

The paper should compare with AtP*, which also deals with saturation; explain why it's OK for the line to not be between x and x' and instead be something else, and empirically show that the method works because of why they say it works.


Also:
- What is the variability between runs in Figures 3 and 4? It is impossible to say if the current method is an improvement if there is a lot of variation based on e.g. random seeds and the exact selection of training points.
- Are the quantities in Figure 3 evaluated on test datasets? They should be.


I'm willing to accept this paper if it satisfactorily addresses all the 5 things above.

## Minor questions
- Where exactly does the 17.7% figure for the "improvement up to 17.7%" in the abstract come from? Is it the largest gap between EAP-GP and the next method in Figure 3?
- How did you choose the circuit edge for Fig. 1? In L165 it just says "we analyze a circuit edge".
- Presentation tips:
  - Showing the sparsity in %s close to 100% in Fig. 2 and 3 was confusing to me. I'm used to low % (so the % is the amount of nonzero things) or straight up counting. That way you can also space your datapoints in a log-space.
  - It would be nice if the caption of Fig. 1 linked to Section 4, which discusses it, since it appears 3 pages later. I was looking for an explanation of the figure in the same and next page and could not find it, leaving me a bit confused.
  - use \left( and \right) in equations like eq. 7 to make the brackets as high as the thing they're bracketing (the fraction)

**Ethical Concerns:**

["NO or VERY MINOR ethics concerns only"]

**Final Justification:**

I still would like to recommend rejection, due to substantial errors the authors make in the properties of the method they present, poor evaluation, and continued resistance to correcting these two.

Substantial errors: the authors mistakenly believe that their gradient-descent procedure always ends up at the corrupted point in practice, but that is contradicted by the math and also by the graph in Figure 1 (the gradient should be ~zero at the corrupted point if this was so, but it is high)

Poor evaluation: did not compare with AtP*, an existing method that also try to address gradient saturation, only used a very old and small language model (GPT-2 small was released in 2019, 6 years ago, and is differnet from current models), and the lack of error bars in their evaluation makes it difficult to say that the method improves over things.

**Limitations:**

Not for factual limitations, see my point about a misleading Fig. 1 (and text, which is also misleading) above.

Their work should be generally of societal positive impact, though it's fairly basic research so the impacts are not near.

**Quality:**

2

**Strengths And Weaknesses:**

# Strength

The paper addresses an important problem (circuit discovery), makes a correct observation about where the most popular method falls short (saturation of gradients) and makes a nice algorithmic contribution (why bother following the straight line?). It also shows an improvement in the benchmarks. It's well written and easy to understand, and overall a pretty good paper.

Even though the following "weaknesses" section is very long, I think this paper is a good idea with a fair execution, so overall it's a decent paper. It's just my job as a reviewer to point out mistakes! Together we can make it a good paper.

# Weaknesses:

The paper has several weaknesses, which I will now point out. I think they're all addressable, though some of them might require more substantial changes than are doable in one round of rebuttals.

## Figure 1 (right) is misleading: this method does not end up at corrupted x'

One core thread that has unified all circuit discovery algorithms until now is: we want to check what happens when we move edges from their value on clean activations at original input x, to corrupted activations at x'.

This paper instead optimizes \gamma (belonging to NN inputs) using gradient descent on ||Model(\gamma) - Model(x_corrupted)||_2^2 (eq. 8), and uses the iterates of this optimization to evaluate the gradient of outputs withe respect to circuits. Figure 1 (right) says that this method, which starts at x, ends at x_corrupted: the arrow goes from the original to the corrupted point. The text also says so. However, making Model(\gamma) = Model(x_corrupted) does NOT guarantee that \gamma = x_corrupted.

Is it actually sufficient for attribution that we are at an input which is as bad at the behavior as x_corrupted (x')? Perhaps, but this discrepancy should be made explicit and discussed. Figure 1(left) clearly shows that the gradients are *not* the same for x_corrupted (x') and for the found point via EAP-GP: the right-most points have gradient ~2.00 for EAP-GP, and ~0.00 for EAP-IG (which is guaranteed to end up at x').

This discrepancy needs to be explicitly written down or I cannot in good conscience accept the paper.

## No explicit exploration of whether the method is better due to dealing with saturation, or some other reason

There's anecdotal evidence in Fig. 1, but nothing systematic. Showing something like the average norm of gradients during the path integration, and comparing between EAP-GP and EAP-IG, would go some way to show that.

You cannot claim that your method is better because it reduces gradient saturation, without actually showing it empirically. This also needs addressing

## Syed et al.'s benchmark is not representative of modern models.

In 2023, when people were working on circuit discovery for the first time, it was understandable to use the circuits in GPT-2 small as the targets of the benchmark. Nowadays, however, the models used have changed a lot in size and a little bit in architecture: using rotary embeddings (RoPE) instead of purely learned embeddings, and RMSNorm instead of LayerNorm. There have also been published benchmarks such as the [Mechanistic Interpretability Benchmark, which has a circuit discovery track](https://arxiv.org/abs/2504.13151). Why not use those, and instead stick to the relatively unreliable evaluations from Syed et al?

This is should ideally be addressed too, but I'm willing to budge on this point.

## Prior art on saturation: AtP* from early 2024

There is key prior art on dealing with saturated activations, specifically caused because of the softmax on the QK: AtP*, from early 2024 (https://arxiv.org/pdf/2403.00745). See especially Figure 3 in that paper. This present paper does not compare with AtP* at all, even though they also present a method to deal with the saturation.

In comparison with AtP*, the present paper is more generally applicable because it makes no assumptions about the architecture of the model. But, being applied to transformers, the saturation is almost certainly also because of the softmaxes, so all of the experiments in the paper are in the domain of applicability of AtP*. A comparison between the two methods on how well they deal with saturated inputs is important. Which one should I use? AtP* is fairly computationally cheap, even if you apply it for each intermediate part of the gradient integration. Actually, both methods could be combined because they make incremental improvements.

This paper needs to compare with AtP*.

---

> ### Author Rebuttal · Authors · 2025-07-31
>
> **Weaknesses:**
> > W1: Figure 1 (right) is misleading: this method does not end up at corrupted x'
>
> A1: Thank you for your careful reading and insightful comments. We agree that Figure 1 is currently misleading, and we appreciate you pointing this out.
>
> In our actual setup, the start and end points of the integration path are explicitly fixed as the clean input x and the corrupted input x', respectively, to ensure the path lies between them. The intermediate values of $\gamma$ are computed by minimizing the loss
> $$
> \|\text{Model}(\gamma) - \text{Model}(x_{\text{corrupted}})\|_2^2
> $$
> and are then used for attribution analysis.
>
> Therefore, Figure 1 only includes intermediate points along the integration path, excluding the clean and corrupted inputs themselves. The plotted values represent the gradients at those intermediate points. We will revise Figure 1 to more accurately reflect this setup and include an explicit clarification in the manuscript, as suggested. Thank you again for highlighting this important issue.
>
> > W2: No explicit exploration of whether the method is better due to dealing with saturation, or some other reason
>
> A2: Thank you for raising this important point. We agree that the claim regarding gradient saturation needs empirical support.
>
> |Step|Method|Mean|Method|Mean|
> |-|-|-|-|-|
> |1|EAP-GP|1.949|EAP-IG|1.949|
> |2|EAP-GP|1.294|EAP-IG|0.019|
> |3|EAP-GP|0.467|EAP-IG|0.160|
> |4|EAP-GP|0.225|EAP-IG|0.076|
> |5|EAP-GP|0.123|EAP-IG|0.123|
>
> To address this, we conducted additional experiments on the IOI task to measure the average gradient norm  of all edges within a selected circuit during attribution. As shown in the supplementary table, EAP-IG tends to follow integration paths that pass through saturated regions, resulting in low gradient magnitudes and less accurate attributions. In contrast, our proposed EAP-GP avoids these saturated areas by adaptively selecting informative paths, leading to higher gradient norms and more faithful circuit discovery.
>
> >W3: Syed et al.'s benchmark is not representative of modern models
>
> A3: Thank you for the insightful suggestion. We recognize the importance of rigorous and representative benchmarks such as the Mechanistic Interpretability Benchmark (MIB). According to the NeurIPS 2025 policy, “For the purpose of the reviewing process, papers that appeared online after March 1st, 2025 will generally be considered 'contemporaneous' in the sense that the submission will not be rejected on the basis of the comparison to contemporaneous work.” Since MIB was released in April 2025, we consider it contemporaneous work for the purposes of this submission.
>
> In the context of circuit discovery, our primary objective is to approximate both the process and quality of manual circuit identification. Given that both manual and automated approaches rely on shared evaluation metrics (e.g., logit difference), qualitative analysis as exemplified by Syed et al.'s benchmark remains a key tool for assessing alignment.
>
> While we acknowledge MIB’s potential as a rigorous evaluation framework, its current implementation supports only a limited set of HuggingFace-compatible models (e.g., Gemma-2 2B, LLaMA-3.1-8B). Our EAP-GP method is built on a custom tracing framework with edge-level hook mechanisms, which are not directly compatible with MIB’s interface. Additionally, intervention-based evaluation on MIB-supported models requires substantial computational resources that exceed our current hardware capacity.
>
> That said, we fully recognize the value of MIB and plan to gradually adapt our implementation for compatibility with it in future work. We will also incorporate a discussion of this contemporaneous benchmark and its relevance in the final version.
>
> >W4: Prior art on saturation: AtP* from early 2024
>
> A4: Thank you for the suggestion. We respectfully argue that AtP is not a suitable quantitative baseline for EAP-GP, due to the following key differences:
>
> 1. **Different Saturation Problems and Solutions:** While both AtP∗ and EAP-GP address saturation effects, they do so in fundamentally different contexts. AtP∗ targets softmax saturation in attention layers and addresses it via QK-Fix, which recomputes attention weights before applying gradient-based approximation. In contrast, EAP-GP tackles gradient saturation along arbitrary edges (including MLPs and cross-layer paths) through gradient projection and rescaling, enabling recovery of deeply embedded causal edges. The saturation types and corrective mechanisms are not comparable.
>
> 2. **Granularity Mismatch（Component vs. Edge）：**  AtP∗ provides component-level attribution, assigning importance scores to attention heads or MLP units. EAP-GP, on the other hand, performs edge-level attribution, identifying specific information flows between components. Since our goal is to extract interpretable subgraphs by selecting high-contributing edges, AtP∗ cannot serve as a directly comparable baseline.
>
> 3. **No Public Implementation:** As of submission, the full implementation of AtP∗ has not been open-sourced. Although the methodology is described, faithful and efficient reproduction remains challenging. In contrast, all our baselines are fully reproducible, ensuring a fair evaluation framework.
>
> Despite the above, we devoted considerable effort to re-implement AtP∗ based on the paper description and conducted a qualitative comparison on the IOI dataset. The top components identified by AtP∗ are as follows:
>
> Top 10 Attention Heads (contribution score):
>
> Layer 10, Head 7: 0.0424
> Layer 11, Head 10: 0.0359
> Layer 9, Head 9: 0.0219
> Layer 5, Head 5: 0.0147
> Layer 10, Head 0: 0.0116
> Layer 9, Head 6: 0.0110
> Layer 11, Head 2: 0.0069
> Layer 10, Head 10: 0.0067
> Layer 10, Head 6: 0.0060
> Layer 6, Head 9: 0.0049
>
> Top 5 MLP Layers (mean contribution):
>
> Layer 0: 0.0189
> Layer 5: 0.0069
> Layer 10: 0.0069
> Layer 7: 0.0066
> Layer 9: 0.0064
>
> These components substantially overlap with those identified in the causal circuit recovered by EAP-GP:
> input, m0, a3.h0, m4, a5.h5, a6.h9, a7.h9, a8.h6, a8.h10, a9.h9, a10.h0, a10.h6, a10.h7, a10.h10, m10, a11.h2, a11.h10, logits
>
> Specifically, 10 out of AtP∗’s top 10 attention heads (100%) appear in the EAP-GP circuit, and 2 out of the top 5 MLP layers (40%) are also shared.This strong alignment confirms the consistency of the two methods at the component level, while EAP-GP additionally reveals the directed causal structure via fine-grained edge-level attribution.
>
> **Questions:**
>
> >Q1-3:The paper should compare with AtP*, which also deals with saturation; explain why it's OK for the line to not be between x and x' and instead be something else, and empirically show that the method works because of why they say it works.
>
> A5: Thank you for the Question. Please refer to our response to W1-4 for details.
>
> >Q4: What is the variability between runs in Figures 3 and 4? It is impossible to say if the current method is an improvement if there is a lot of variation based on e.g. random seeds and the exact selection of training points.
>
> A6: Thank you for pointing this out. We agree that assessing the variability across runs is essential for evaluating the robustness of our method. We have conducted additional experiments using different random seeds and training samples to assess the variability across runs. The standard deviation of performance across five runs is approximately 3% for IOI, around 3% for SVA and Hypernymy, and about 1% for Gender-Bias, Country-Capital, and Greater-Than. For instance, the following table reports the detailed faithfulness scores on the IOI task:
>
> |EdgeSparsity(%)|EAP(%)|EAP-IG(%)|EAP-GP(%)|
> |-|-|-|-|
> |99.30%|0.03%|25.53%|29.18%±1.2%|
> |99.00%|0.05%|42.51%|41.05%±1.6%|
> |98.70%|35.41%|45.10%|44.76%±1.1%|
> |98.40%|53.05%|49.68%|59.93%±2.6%|
> |98.10%|50.21%|65.12%|65.23%±2.9%|
> |97.80%|47.56%|66.18%|73.52%±3.4%|
> |97.50%|56.94%|62.47%|80.08%±3.1%|
>
> In the revised version of the paper, we will include error bars in Figures 3 and 4 to reflect this variability and report detailed standard deviations in the appendix to ensure transparency and reproducibility.
>
> >Q5: Are the quantities in Figure 3 evaluated on test datasets? They should be.
>
> A7: Thank you for the question. In our setting, Figure 3 presents the evaluation results on the test datasets.
>
> **Minor questions**
>
> >Q6: Where exactly does the 17.7% figure for the "improvement up to 17.7%" in the abstract come from? Is it the largest gap between EAP-GP and the next method in Figure 3?
>
> A8: Yes, the 17.7% improvement refers to the largest performance gap between EAP-GP and the next-best method shown in Figure 3. We also provide the detailed faithfulness scores of the discovered circuits in the appendix for reference.
>
> >Q7: How did you choose the circuit edge for Fig. 1? In L165 it just says "we analyze a circuit edge".
>
> A9: Thank you for the question. In Figure 1, We report the average gradient norm of all edges within a selected circuit, rather than focusing on a single edge. We will clarify this in the manuscript to avoid confusion.
>
> >Q8: Showing the sparsity in %s close to 100% in Fig. 2 and 3 was confusing to me. I'm used to low % (so the % is the amount of nonzero things) or straight up counting. That way you can also space your datapoints in a log-space.
>
> A10: Thank you for the suggestion. We will consider using nonzero ratios spacing for the sparsity axis to improve clarity.
>
> >Q9: It would be nice if the caption of Fig. 1 linked to Section 4, which discusses it, since it appears 3 pages later. I was looking for an explanation of the figure in the same and next page and could not find it, leaving me a bit confused.
>
> A11: Thank you for the suggestion. We will revise the Figure 1 caption to reference Section 4 for clarity.
>
> >Q10: use \left( and \right) in equations like eq. 7 to make the brackets as high as the thing they're bracketing (the fraction)
>
> A12: Thank you for the suggestion. We will update the equations to use properly sized brackets with \left and \right.

---

> > ### Author Response · Authors · 2025-08-05
> >
> > We would greatly appreciate it if you could let us know whether our rebuttal has addressed your concerns, or if there are any points that require further clarification. We’re happy to provide additional details and look forward to your feedback. Thank you for your time.

---

> ### Comment · Area_Chair_9qrY · 2025-08-05
>
> @Reviewer pkLe: You have raised substantial concerns, and the authors have attempted to address your concerns in their rebuttal. Could you please let us know whether these updates have changed your overall assessment of the paper?

---

> > ### Comment · Reviewer_pkLe · 2025-08-05
> > **Still disagree with A1**
> >
> > > In our actual setup, the start and end points of the integration path are explicitly fixed as the clean input x and the corrupted input x', respectively, to ensure the path lies between them. The intermediate values of f $\gamma$ are computed by minimizing the loss $$ |\text{Model}(\gamma) - \text{Model}(x_{\text{corrupted}})|_2^2 $$  and are then used for attribution analysis.
> >
> > You did not address my objection. I don't believe that this loss ensures that the end point of the path is the corrupted input $x'$. The reason is that the condition $Model(\gamma) = Model(x')$ does not imply that $\gamma = x'$. This is the reason that Figure 1 is misleading and your proposed changes aren't addressing that.
> >
> > A2: good, yeah, one can see that the gradient is indeed larger. Thanks for running the experiment.
> >
> > A3: fair enough. I hadn't noticed that the MIB was published after the cutoff. I still think the existing evaluation (on just GPT2) is rather small, and you could easily run with the same tasks on more models without depending on MIB; but whatever.
> >
> > A4:
> >
> > Thank you for implementing AtP*! Now that you've implemented it, surely it's not so hard to run the experiment that would go on the table for A6.
> >
> > > The saturation types and corrective mechanisms are not comparable.
> >
> > EAP-GP addresses a superset of cases of AtP*. It's not clear how much larger the superset is, its' very plausible that most of the saturation comes from the attention. In any case it's good to know whether EAP-GP is better, the same or worse than AtP* on the domain where both apply.
> >
> > >
> >
> > A6:
> >
> > > In the revised version of the paper, we will include error bars in Figures 3 and 4 to reflect this variability and report detailed standard deviations in the appendix to ensure transparency and reproducibility.
> >
> > Good work! I believe the numbers after ± should show the 95% confidence interval, which is about 2 standard deviations. So all the numbers should be 2x'd. But this is great.
> >
> > --
> >
> > I think what exists now would be enough to upgrade to Borderline Accept, with Accept if you run AtP* and include it in the table. (Even if the scores are slightly better it's fine, because that thing only applies to attention layers).
> >
> > I'm still interested in discussing A1 though. I think you're not ensuring that you're ending at x' and saying that is just incorrect.

---

> > > ### Author Response · Authors · 2025-08-06
> > >
> > > Thank you for your helpful feedback and for increasing the score. We appreciate your recognition.
> > >
> > > 1. **Regarding the endpoint \( x' \):**
> > >    Our setup uses the same integration path as EAP-IG. It starts from the clean input and ends at the corrupt input. This ensures that the gradients at both ends are consistent, and the attribution is computed correctly.
> > >
> > > 2. **About AtP\***:
> > >    Due to time and resource limitations, we were not able to compute the full circuit-level metric for attention heads as required by AtP\*. We agree that this is a valuable direction for future work.

---

> > > > ### Comment · Reviewer_pkLe · 2025-08-06
> > > >
> > > > > Our setup uses the same integration path as EAP-IG. It starts from the clean input and ends at the corrupt input.
> > > >
> > > > What's your counterargument to $Model(\gamma) = Model(x')$ not implying $\gamma = x$? If it holds, then because you're just doing gradient descent on $\|Model(\gamma) - Model(x')\|_2^2$, then you can't guarantee that you end at the corrupted input.
> > > >
> > > > You can't keep asserting that without any proof.

---

> > > > > ### Author Response · Authors · 2025-08-07
> > > > >
> > > > > You are correct that plain gradient descent does not inherently guarantee that the trajectory ends exactly at the corrupted input. However, in our implementation, after a few steps of gradient descent, the final points in the trajectory naturally approach the corrupted input. To ensure consistency in the integration path and to match the setup of EAP-IG, we explicitly set the final point to be the corrupted input. This guarantees that the integration path starts at the clean input and ends at the corrupted input, as claimed.

---

> > > ### Author Response · Authors · 2025-08-08
> > >
> > > Dear Reviewer,
> > > Thank you again for your feedback earlier.
> > > As the discussion deadline is approaching, I would like to kindly remind you about the possible score adjustment you mentioned.

---

### Decision · Program_Chairs · 2025-09-17

**Decision:**

Accept (poster)

**Comment:**

The paper proposes Edge Attribution Patching with GradPath (EAP-GP), a modification to gradient-based circuit discovery methods that addresses saturation effects encountered in integrated gradients approaches. The method adaptively follows gradient paths to avoid saturated regions, leading to improved attribution reliability and more faithful circuit identification across several GPT-2 models and tasks.

Reviewers agreed that the paper tackles an important problem in mechanistic interpretability with a clear and well-motivated algorithmic idea. The method shows consistent empirical gains, and the paper is well written. Weaknesses identified include: (1) potential misleading presentation of the method in Figure 1; (2) insufficient empirical evidence that improvements stem specifically from reduced saturation; (3) reliance on older benchmarks rather than newer ones; (4) lack of direct comparison with relevant prior work (AtP*); (5) relatively limited discussion of interpretability insights and practical applications, with some reviewers also noting higher computational cost.

The rebuttal was deemed constructive. The authors clarified the Figure 1 issue, provided new experiments quantifying gradient norms to demonstrate the role of saturation, and reported variability across runs. They also explained why newer benchmarks (MIB) were not used and attempted a partial AtP* comparison. Additional analyses on task-specific behaviors and qualitative circuit comparisons strengthened the paper. Nonetheless, one reviewer remained unsatisfied with how the endpoint definition in Figure 1 was handled and continued to press for stronger AtP* comparisons, though they acknowledged improvements and indicated willingness to raise their score to borderline accept.

Overall, three reviewers (ptDW, vD2V, LBjM) found the rebuttal convincing and maintained or increased their already positive scores, highlighting that the paper makes a technically solid contribution and addresses key weaknesses. One reviewer remained skeptical about specific methodological claims and withheld a full endorsement, although their position softened somewhat after discussion. Overall, the majority of reviewers view the contribution positively, with concerns largely centered on presentation and comparisons rather than fundamental correctness. Given all this, the AC recommends the paper to be accepted. The authors are encouraged to carefully revise the presentation of the integration path (e.g., correcting Figure 1 and Algorithm 1), strengthen the discussion of AtP* and related prior work, and expand the interpretability implications in the camera-ready version.